# A vaccine central in A(H5) influenza antigenic space confers broad immunity

Adinda Kok[1,4], Samuel H. Wilks[2,5,8], Sina Tureli[2,8], Sarah L. James[2,8], Theo M. Bestebroer[1], David F. Burke[2,6], Mathis Funk[1,9], Stefan van der Vliet[1], Monique I. Spronken[1], Willemijn F. Rijnink[1], David J. Pattinson[2,7], Dennis de Meulder[1], Miruna E. Rosu[1], Pascal Lexmond[1], Judith M. A. van den Brand[3], Sander Herfst[1], Derek J. Smith[2], Ron A. M. Fouchier[1] & Mathilde Richard[1✉]

Highly pathogenic avian influenza A(H5) viruses globally impact wild and domestic birds, and have caused severe infections in mammals, including humans, underscoring their pandemic potential[1–5]. The antigenic evolution of the A(H5) haemagglutinin (HA) poses challenges for pandemic preparedness and vaccine design[6]. Here the global antigenic evolution of the A(H5) HA was captured in a high-resolution antigenic map. The map was used to design immunogenic and antigenically central vaccine HA antigens, eliciting antibody responses that broadly cover the A(H5) antigenic space. In ferrets, a central antigen protected as well as homologous vaccines against heterologous infection with two antigenically distinct viruses. This work showcases the rational design of subtype-wide influenza A(H5) pre-pandemic vaccines and demonstrates the value of antigenic maps for the evaluation of vaccine-induced immune responses through antibody profiles.

Influenza A viruses are enzootic in wild migratory birds of aquatic habitats around the world[7]. In wild waterfowl, 17 subtypes of HA (H1–H16, H19) and 9 of neuraminidase (NA, N1–N9), surface glycoproteins of influenza A viruses, have been identified[8,9]. HA is the receptor-binding and fusion protein of influenza A viruses, and its head domain contains dominant epitopes targeted by antibodies. The host cell receptors of avian and human influenza viruses are α2,3-linked and α2,6-linked sialic acids, respectively, and receptor binding specificity is one of the major host range determinants[8]. Viruses of the A(H5) and A(H7) subtypes are particularly important, as they can evolve into highly pathogenic avian influenza viruses (HPAIVs) in poultry, causing severe disease and high mortality. A(H5) HPAIVs from the A/goose/Guangdong/1/1996 (GsGd) lineage were first detected in Hong-Kong in 1997[10,11], and have since spread globally[1,12,13]. This lineage is now enzootic in poultry and wild birds in many regions[1]. A(H5) GsGd HPAIVs have infected over 60 mammalian species[2], caused mass die-offs in marine mammals and widespread outbreaks in US dairy cows[14], leading to substantial animal health and economic impacts. Moreover, the spillover to humans has resulted in severe cases, with 530 fatalities among 1,085 confirmed cases[3–5], raising global pandemic concerns. Continuous virus circulation in birds has led to the genetic and antigenic diversification of the GsGd HPAIVs HA. GsGd HPAIVs belonging to genetic clades 2.3.2.1 and 2.3.4.4 are currently dominant[1]. This antigenic diversity has prompted the World Health Organization (WHO) to biannually select multiple candidate vaccine viruses (CVVs)[6]. Since 2006, 48 candidates have been recommended, highlighting challenges posed by the antigenically diverse GsGd HPAIVs to vaccine design and pandemic preparedness. While necessary, continuous CVVs updating is a reactive and unsustainable approach to prepare against A(H5) HPAIVs that might emerge in the future. At present, manufacturing an inactivated vaccine that antigenically matches the pandemic virus remains the gold standard, which usually takes up to 6 months, leaving populations vulnerable to infection. Such vaccines generally have low immunogenicity, owing to the need to induce a primary immune response in an immunologically naive population, and to the intrinsic low immunogenicity of avian influenza viruses[15]. Although alternative vaccine platforms that may offer higher immunogenicity are under investigation, such as mRNA and vector-based approaches, the choice of vaccine antigen remains a crucial component of vaccine effectiveness.

An integrated analysis of the global antigenic evolution and historical diversification of the A(H5) GsGd HPAIVs is missing, yet is crucial to design vaccine antigens that induce immunity against antigenically distinct viruses. We aimed to characterize this evolution by creating a high-resolution antigenic map, using a large collection of historical, recent and current A(H5) influenza viruses. This map was used to design immunogenic and antigenically central vaccine antigens conferring broad reactivity to antigenically distinct viruses, and to visualize antibody-mediated immune responses. A proof of concept of high immunogenicity, broad reactivity and protection is provided using ferrets as a preclinical model.

[1]Department of Viroscience, Erasmus University Medical Center, Rotterdam, The Netherlands. [2]Center for Pathogen Evolution, University of Cambridge, Cambridge, UK. [3]Division of Pathology, Faculty of Veterinary Medicine, Utrecht University, Utrecht, The Netherlands. [4]Present address: Biomolecular Mass Spectrometry and Proteomics, Bijvoet Center for Biomolecular Research, Department of Chemistry, Faculty of Science, Utrecht University, Utrecht, The Netherlands. [5]Present address: Institute for Electrical and Electronic Engineering, Faculty of Engineering and Information Technology, University of Melbourne, Melbourne, Victoria, Australia. [6]Present address: Centre for Host-Microbiome Interactions, King's College, London Tower Wing, Guy's Hospital, London, UK. [7]Present address: Influenza Research Institute, Department of Pathobiological Sciences, School of Veterinary Medicine, University of Wisconsin-Madison, Madison, WI, USA. [8]These authors contributed equally: Samuel H. Wilks, Sina Tureli, Sarah L. James. [9]Deceased: Mathis Funk. ✉e-mail: m.richard@erasmusmc.nl

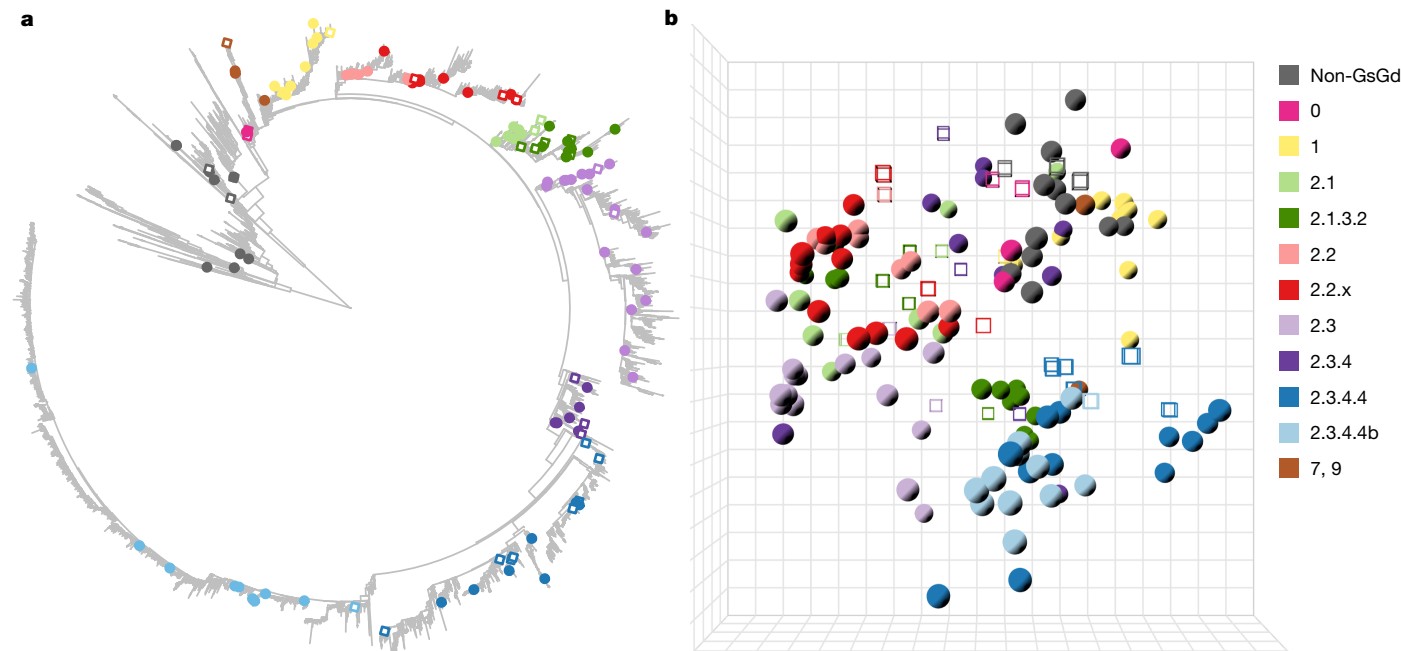

**Fig. 1 | High-resolution three-dimensional A(H5) antigenic map comprising genetically diverse HAs. a**, Maximum-likelihood phylogenetic tree based on 14,896 A(H5) nucleotide sequences, rooted at the midpoint, corresponding to the divergence of the Eurasian and American non-GsGd lineages. The HAs selected for antigenic characterization are highlighted with closed circles or open squares, colour coded based on their respective genetic lineage, as indicated in **b**. The open squares represent HAs of viruses against which homologous ferret sera were raised. A zoomable PDF file showing the isolate names is provided in Supplementary Data 1 and an interactive version of the tree is available online (https://itol.embl.de/tree/15683116022254171827999907).

**b**, The three-dimensional antigenic map constructed from HI data of 117 antigens titrated against 29 post-infection sera. Antigens are represented as closed spheres and sera as open cubes. Antigens and sera are colour coded on the basis of their respective genetic lineage, as shown on the right. Each direction (x, y, z) represents antigenic distance, and one square of the grid corresponds to one antigenic unit, defined as a twofold change in HI titre. The corresponding interactive display (Supplementary Data 2) provides additional information, including visualization of the map from different angles, and display of antigen and serum names.

## A(H5) influenza virus antigenic map

We selected 127 representative HA genes of (sub)clades along the phylogenetic tree of A(H5) influenza viruses as antigens (Fig. 1a, Supplementary Table 1 and Supplementary Data 1; all supplementary data files are also available at https://epiv-lab.github.io/H5-antigenically-central-vaccine/). HAs from non-GsGd viruses from the American and Eurasian lineages, and HAs of WHO CVVs or closely related viruses were also included (Fig. 1a and Supplementary Table 2). For viruses that were not available in-house, synthetic HA genes were ordered and cloned into plasmids to produce recombinant viruses in the genetic background of the attenuated strain A/Puerto Rico/8/1934 (PR/8) using reverse genetics. Ferret sera were generated against 33 of these viruses based on their divergent genetic and antigenic properties as assessed in preliminary assays (Fig. 1a and Supplementary Table 3). The resulting dataset comprised 127 antigens and 33 post-infection sera, which were cross-titrated in haemagglutination inhibition (HI) assays (Supplementary Table 4 and Supplementary Note 1). Antigenic cartography, a tool developed to quantitatively interpret large cross-HI datasets[16], was then used to visualize and evaluate the antigenic evolution and diversification of A(H5) influenza A viruses from 1959 to 2022 (Fig. 1b and Supplementary Data 2). An antigenic map was constructed using multidimensional scaling algorithms described previously[16]. Distances in antigenic maps are inversely correlated with HI titres; that is, antigens are close in space with sera against which they react with a high HI titre. Antigenic maps also enable the visualization of the antigenic relatedness between antigens, which is not directly measured in the HI assay.

The generation and validation of the antigenic map are described in Supplementary Notes 2 and 3 (Supplementary Data 3 and 4, and

Extended Data Figs. 1–4). Minimally, three dimensions were required to represent the A(H5) HI titre data. Ten antigens showed reactivity with fewer than four ferret sera and were removed, along with their homologous sera, as less than four detectable titres is insufficient to confidently place points in a map of three or more dimensions (Supplementary Table 2 and Supplementary Note 1–3). The resulting map contained 117 antigens and 29 post-infection sera. The three-dimensional antigenic map represented the underlying HI data well, as shown by the correlation between the antigen/serum pairwise distances from HI titres and map Euclidean distances ($R^2 = 0.64$; Extended Data Fig. 3a). Additional analyses were performed to evaluate the map's accuracy and stability (Supplementary Note 3, Supplementary Data 3 and 4, and Extended Data Figs. 3 and 4). Moreover, a genetically and antigenically diverse representative subset was used to perform virus neutralization assays, and virus neutralization titres were shown to correlate with the HI titres ($R^2 = 0.61$; Extended Data Fig. 3d).

Antigenic evolution of A(H5) influenza viruses showed a non-directional pattern over time (Supplementary Video). Analyses of the genetic and antigenic distances revealed a relative discordance between genotype and antigenic phenotype (Extended Data Fig. 5a). For example, antigens from different genetic (sub)clades were, on some occasions, close in antigenic space (Fig. 1b and Supplementary Data 2), suggesting similar antigenic properties. To assess whether this might have partially resulted from map distance distortions due to dimension constraints, we compared raw HI titres for antigen pairs with a large pairwise genetic distance (more than 25 amino acid differences in HA1, the domain which contains the HA head), and a low pairwise antigenic distance (below 1.5 antigenic units (AU), where one AU corresponds to a twofold change in HI titre). For 19 out of 24 pairs, similarity in HI reactivity was indeed observed (Extended Data Fig. 6;

$R^2 > 0.5$), providing evidence that antigenic similarity and not map distortion accounted for these observations.

The maximum pairwise distance between the non-GsGd antigens, spanning 43 years, was only 5.58 AU. By contrast, the GsGd antigens, spanning 25 years, were positioned at a maximum pairwise distance of 14.71 AU. This highlights the increased level of antigenic evolution relative to the timespan in the GsGd lineage as compared with that of the non-GsGd antigens (Extended Data Fig. 5b–d and Supplementary Video). Within the GsGd lineage, little antigenic evolution was observed for clade 1 and 2.2 (Supplementary Table 5), while clades 2.1 and 2.3 were more divergent, with maximum distances of 10.15 and 14.71 AU between antigens, respectively. Antigenic diversity was also observed within the 2.3.2.1 and 2.3.4.4 clades, with maximum pairwise distances of 7.37, 8.780 and 5.7251 AU between clades 2.3.2.1, 2.3.4.4 and 2.3.4.4b antigens, respectively. Moreover, antigenic differences were noted between clades 2.3.2.1 and 2.3.4.4 (Supplementary Table 5), with a maximum pairwise distance of 11.88 AU, and of 6.84 AU between their most recent antigens. These observations underscore the challenge posed by the antigenic diversity of GsGd A(H5) viruses for pandemic preparedness, including vaccine design. To address these challenges, the WHO has selected 48 CVVs over the years[6]. Highlighting the A(H5) WHO CVV(-like) antigens (Supplementary Table 2) present in our dataset in the antigenic map revealed a good coverage of the antigenic space (Supplementary Data 5a), but frequent updates are required to cope with antigenic evolution.

## Design of a central vaccine antigen

An alternative to the continuous production of CVVs is the selection or design of antigen(s) inducing cross-reactive immune responses. Theoretically, high cross-reactivity could be achieved in a non-directional space with a single antigen, provided that it (1) elicits an immune response centred in antigenic space and (2) is highly immunogenic, that is, it induces a high homologous HI titre. Notably, the maximum distance of antigens from the centre of the map has not increased substantially since 2010, despite ongoing evolution (Extended Data Fig. 7a), suggesting that the A(H5) antigenic space might be constrained. This observation underscores the feasibility and sustainability of designing immunogenic and antigenically central vaccine antigens.

To design such an antigen, our initial exploration focused on wild-type antigens within 2.5 AU of the antigenic map's centre, defined as the arithmetic mean of all antigens positions. Twelve antigens qualified (Supplementary Table 2 ('notes' column)), which reacted with a higher number of sera and with a higher geometric mean titre (GMT) than those further away from the centre (Extended Data Fig. 7b,c). Eleven of the twelve lacked a putative glycosylation site at position 154, a feature that is present in 37% of the map's GsGd antigens. The removal of putative HA glycosylation sites has been documented to enhance vaccine immunogenicity[17,18]. Moreover, several central antigens contained amino acids such as Asn94, Ala156, Arg189, Ile210 and Asn223, previously associated with alterations in the HA receptor binding profile[19,20]. These observations, coupled with evidence from the literature highlighting the impact of receptor-binding amino acid changes on increased HA cross-reactivity and immunogenicity[17,21], prompted us to investigate the receptor binding profile of these 12 centrally located antigens. Notably, 8 out of the 12 central antigens exhibited a certain level of binding to α2,6-linked sialic acids (Supplementary Table 6).

Building on these insights and the expectation that the next pandemic influenza virus will probably bind to α2,6-linked sialic acids, we rationally designed candidate vaccine antigens (CVAs) based on non-central HAs of viruses from distinct genetic clades. Substitutions were introduced at positions altering the receptor-binding profile (Q222L and G224S)[22] and removing a putative glycosylation site at position 154 (T156A), resulting in CVA-Vietnam, based on A/Vietnam/1194/2004 (clade 1), CVA-Indonesia, based on A/Indonesia/5/2005 (clade 2.1.3.2),

and CVA-Anhui, based on A/Anhui/1/2005 (clade 2.3.4). These antigens exhibited a dual receptor binding profile to both α2,3- and α2,6-linked sialic acids, whereas their parent HAs exclusively bound to α2,3-linked sialic acids (Supplementary Table 6).

These antigens were tested in HI assays against all sera from the dataset (Supplementary Table 7), and were positioned in the antigenic map. CVAs exhibited increased HI reactivity and were situated on average 1.33 AU from the map centre, 4.15 AU closer than their respective wild-type counterparts. The position of the CVAs in antigenic space served as an indication of their cross-reactive potential. However, more relevant was the induction of an immune response that is both centred in space and immunogenic. To assess these parameters, two ferrets were vaccinated and boosted with Addavax-adjuvanted whole-inactivated vaccines, 28 days apart. As a comparator, one of the natural central antigens mentioned earlier, A/Iraq/755/2006, was used (Iraq). Antibody profiles were generated from sera obtained 4 weeks after boost (Iraq$_{VACC}$, CVA-Vietnam$_{VACC}$, CVA-Indo$_{VACC}$, CVA-Anhui$_{VACC}$) by HI titration against 113 antigens (Fig. 2 and Supplementary Data 6; see Supplementary Data 7 and Supplementary Table 8 for data from individual animals). Except for one Iraq$_{VACC}$ serum, all sera had detectable titres against over half of the tested antigens. Although CVA-Anhui$_{VACC}$ was slightly less central than CVA-Indonesia$_{VACC}$ (1.60 versus 1.02 AU from the centre, respectively), CVA-Anhui induced the broadest HI response, characterized by the highest GMT (43) against all A(H5) antigens, and the highest GMT against viruses from currently circulating genetic clades 2.3.2.1 and 2.3.4.4 (29 versus 17 for CVA-Indonesia). CVA-Anhui was then selected for further investigation due to its distinct characteristics.

Assessment of the genetic stability of the CVA-Anhui virus through 5 to 10 serial passages in embryonated chicken eggs, the substrate most frequently used to produce influenza virus vaccines, revealed the selection of amino acid substitution T134A in HA in four out of five independent experiments. When this substitution was introduced in CVA-Anhui, it stabilized the virus across two additional experiments. Given that Ala134 is consistently found in all antigens in the antigenic map and is also present in two database entries of the A/Anhui/1/2005 HA (Supplementary Table 2), it was incorporated in the CVA-Anhui antigen (A/Anhui/1/2005$_{T134A,T156A,Q222L,G224S}$). HI assays comparing the resulting CVA-Anhui antigen with the earlier version (without T134A) revealed that the updated antigen, hereafter AC-Anhui, was closer to the antigenic map centre (1.29 AU versus 2.11 AU). AC-Anhui was therefore advanced to preclinical evaluation in ferrets.

## Broad protection by AC-Anhui$_{VACC}$

Preclinical studies in ferrets were performed to assess (1) the protective capacity of AC-Anhui$_{VACC}$ as compared with the standard of care, that is, vaccines homologous to challenge viruses; and (2) the impact of the introduced substitutions on the height and breadth of the HI antibody response, and protection against challenge.

The selected challenge viruses were A(H5N1) clade 2.2.1.2 A/duck/Giza/15292S/2015 (H5N1$_{Giza}$) and A(H5N6) clade 2.3.4.4a A/Sichuan/26221/2014 (H5N6$_{Sichuan}$). These were chosen because their respective HAs are genetically and antigenically distinct from one another (9.19 AU apart), and from the non-central A/Anhui/1/2005 virus (7.40 and 10.61 AU, respectively), the wild-type counterpart of AC-Anhui (Supplementary Data 5b and Supplementary Table 5). Groups of six ferrets were vaccinated twice, 28 days apart, with Addavax-adjuvanted split-inactivated vaccines containing either the antigenically central HA (AC-Anhui$_{VACC}$), the wild-type counterpart HA (Anhui$_{VACC}$), the challenge viruses' HAs (Giza$_{VACC}$ and Sichuan$_{VACC}$) or phosphate-buffered saline (PBS) (Mock$_{VACC}$) (Extended Data Fig. 8). The Giza$_{VACC}$ and Sichuan$_{VACC}$ groups represent the standard of care, that is, vaccines homologous to the challenge viruses, and were used as positive controls. The Anhui$_{VACC}$ group was used as a non-central vaccine antigen control (5.05 AU from

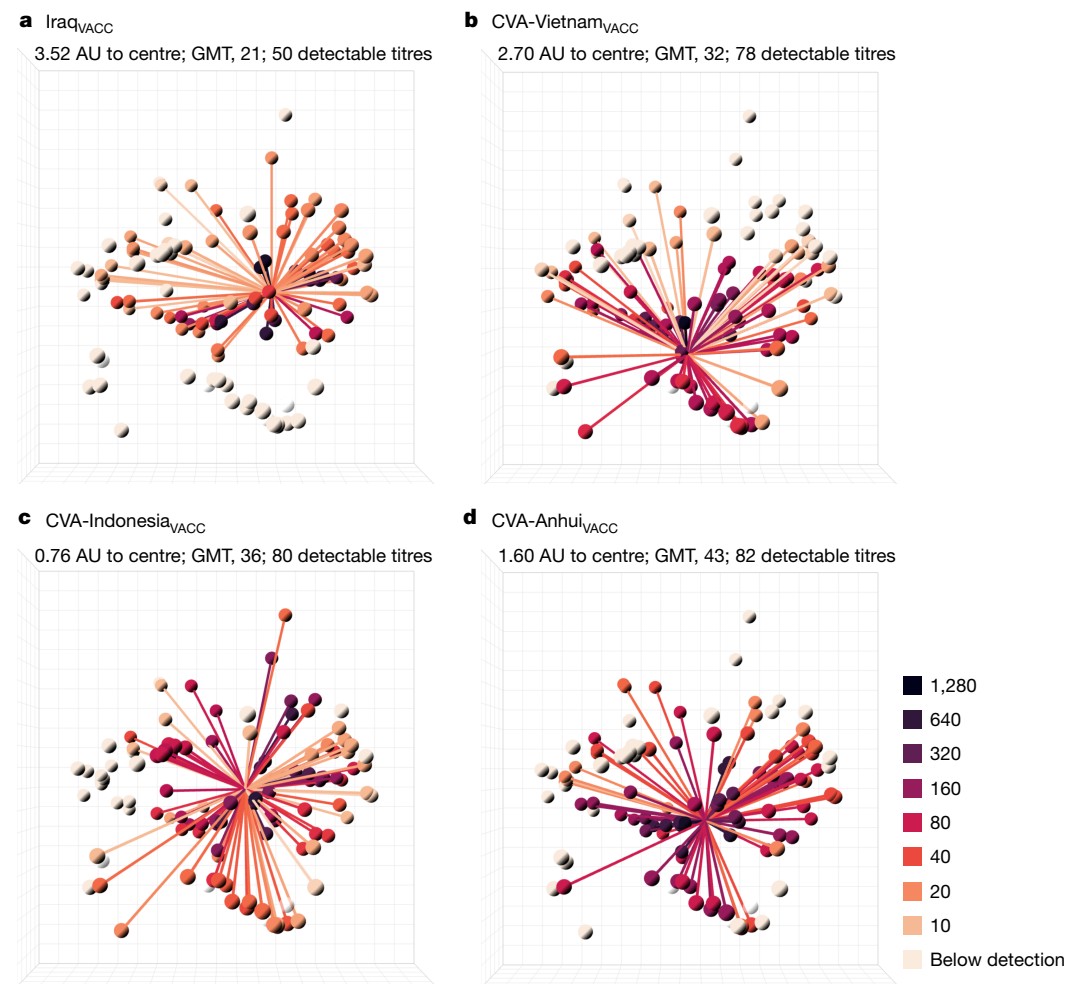

**a** Iraq_VACC

3.52 AU to centre; GMT, 21; 50 detectable titres

**b** CVA-Vietnam_VACC

2.70 AU to centre; GMT, 32; 78 detectable titres

**c** CVA-Indonesia_VACC

0.76 AU to centre; GMT, 36; 80 detectable titres

**d** CVA-Anhui_VACC

1.60 AU to centre; GMT, 43; 82 detectable titres

1,280
640
320
160
80
40
20
10
Below detection

**Fig. 2 | Antibody profiles illustrate the broad cross-reactivity in A(H5) antigenic space of ferret sera raised by vaccination with whole-inactivated vaccines containing mutated HAs. a–d**, For each vaccine, Iraq_VACC (**a**), CVA-Vietnam_VACC (**b**), CVA-Indonesia_VACC (**c**) and CVA-Anhui_VACC (**d**), antibody profiles are displayed in the antigenic map from Fig. 1b. Each plot displays the mean titres for two vaccinated ferrets. The lines connect the position of the serum with the antigens against which a titre above the assay's detection limit was observed (that is, >5 for mean serum, ≥10 for individual serum). Antigens and connecting lines are coloured by the HI titre as indicated on the right. The numbers at the top of each panel indicate the distance of the mean serum position to the centre of the antigenic map in antigenic units (AU), the overall geometric mean titre (GMT) of the mean serum and the group mean of the number of titres above the detection limit per serum. The four antigens that were not titrated against these sera are shown in transparent white. One CVA-Anhui_VACC-vaccinated ferret reached humane end points during the study because of a malignant lymphoma, which was unrelated to the experimental procedures. Owing to the premature euthanasia of this ferret, data of a single serum were used for analysis and visualization in **d**.

the map centre) and to assess the impact of the introduced substitutions in AC-Anhui_VACC. The Mock_VACC group was used as a negative control to fully appreciate disease severity after challenge. We chose to perform preclinical work using split-inactivated vaccines, given that licensed A(H5) pre-pandemic vaccines for human use are almost all of this type[23]. We used an MF-59 like adjuvant, Addavax, as A(H5) influenza vaccines are unlikely to be immunogenic in naive individuals if unadjuvanted[15]. To isolate the effects of varying the HA antigen, the NA in the vaccine was mismatched with the challenge virus NA (Extended Data Fig. 8).

After boosting, an increase in HI titre against the vaccine antigens was observed for almost all vaccinated animals (Supplementary Table 9). The post-boost sera were titrated against 113 antigens of the antigenic map, and these HI data were used to generate mean antibody profiles for each experimental group (Fig. 3 and Supplementary Data 8; see Supplementary Data 9 and 10 and Supplementary Table 8 for data from individual animals). The mean AC-Anhui_VACC post-boost serum was positioned in closer proximity to the centre of the antigenic map than that raised after vaccination with the non-central wild-type antigens (Fig. 3 and Supplementary Data 8). Moreover, the GMT and mean number of antigens with a detectable HI titre were higher for the mean

AC-Anhui_VACC post-boost serum (Fig. 3b,e) than for the mean Anhui_VACC post-boost serum (Fig. 3a,d). These observations indicated that the introduced substitutions led to a more central and enhanced immune response, both in terms of height and breadth. Notably, the height and breath of immune responses in both the Anhui_VACC and AC-Anhui_VACC groups from the H5N1_Giza study exceeded those from the H5N6_Sichuan study, potentially attributable to the use of a different batch of ferrets and vaccines. Anhui_VACC did not induce detectable HI titres against the H5N1_Giza and H5N6_Sichuan challenge viruses in any of the animals (Supplementary Tables 8 and 9). By contrast, AC-Anhui_VACC induced detectable HI titres against the challenge viruses in all ferrets of the H5N1_Giza challenge and four out of six ferrets of the H5N6_Sichuan challenge, showing a comparable response to that observed for the homologous vaccines (Supplementary Tables 8 and 9).

Four weeks after boosting, ferrets were inoculated intranasally and intratracheally with the H5N1_Giza virus or the H5N6_Sichuan virus, at doses determined in preliminary studies to induce a reproducible and consistent upper and lower respiratory tract infection ($10^{5.5}$ and $10^{3.4}$ 50% tissue culture infectious dose (TCID$_{50}$), respectively). Nose and throat swabs were collected daily, alongside monitoring of body weight, body

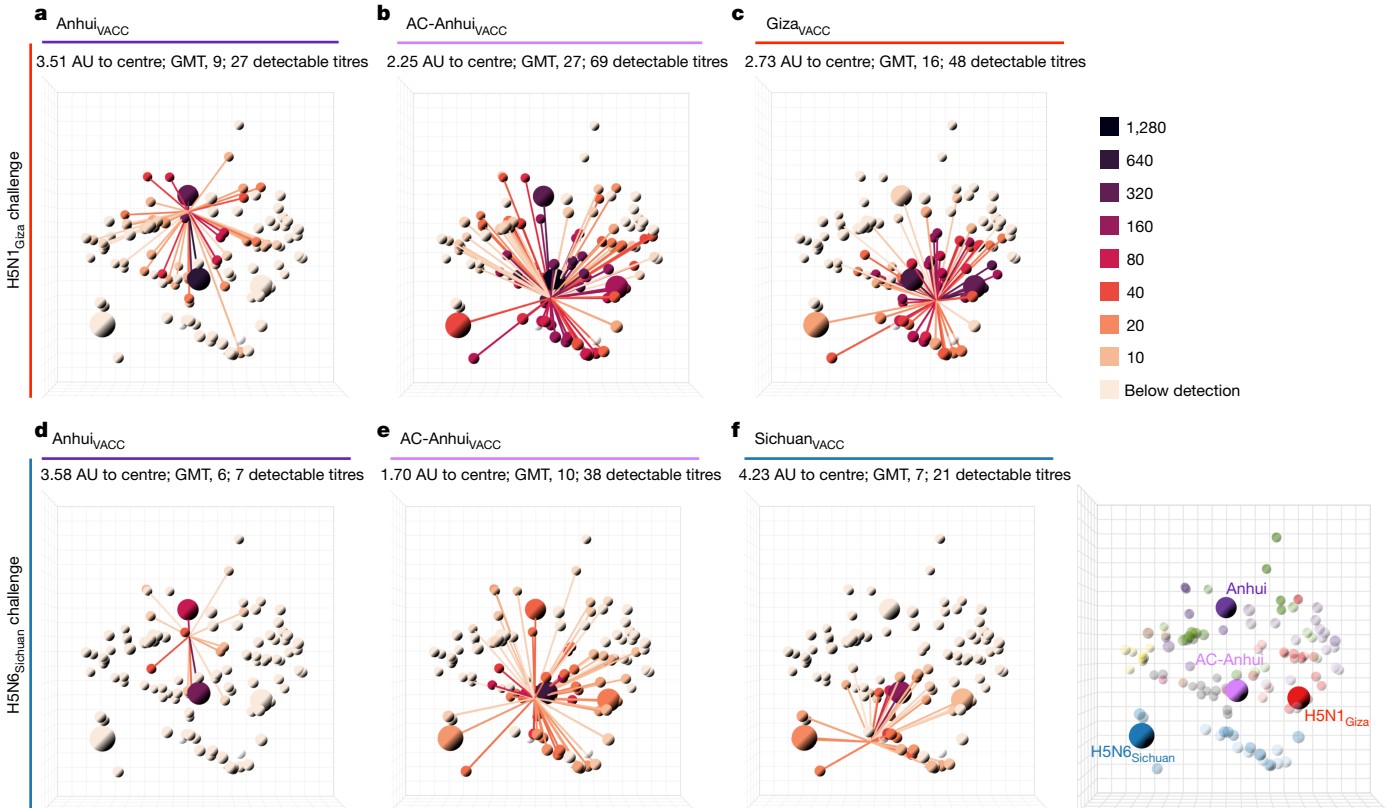

**Fig. 3 | Broader, higher and more-central antibody responses in ferrets after vaccination with split-inactivated vaccines containing a mutated HA as compared with its wild-type counterpart. a–f,** For each vaccine group, antibody profiles representing the mean HI titres per group ($n = 6$) are displayed in the antigenic map from Supplementary Data 5b. Antibody profiles of ferrets from the vaccination–challenge study, challenged with the H5N1$_{Giza}$ (**a–c**) and the H5N6$_{Sichuan}$ (**d–f**) viruses, and vaccinated with vaccines containing the following HA: Anhui$_{VACC}$ (**a** and **d**), AC-Anhui$_{VACC}$ (**b** and **e**), Giza$_{VACC}$ (**c**) and Sichuan$_{VACC}$ (**f**). The same representation as in Fig. 2, with the antigens used in the vaccination–challenge studies highlighted as larger spheres. The antigenic map from Supplementary Data 5b is displayed on the right as a reference. An interactive version of this figure is provided in Supplementary Data 8.

temperature, clinical signs and activity (Extended Data Fig. 8). At 4 days post-inoculation (d.p.i.), ferrets were euthanized and relevant tissues, selected on the basis of virus detection in preliminary studies, were collected for virological and histopathological analyses (Extended Data Fig. 8).

After H5N1$_{Giza}$ virus inoculation, Mock$_{VACC}$ ferrets showed a maximum mean body weight reduction of 11.4% (Fig. 4a), and reduced activity from 3 d.p.i. onwards (Extended Data Fig. 9a), accompanied by an increased breathing frequency at 4 d.p.i. By contrast, vaccinated ferrets showed no substantial alteration in body weight, activity level nor clinical status (Fig. 4a and Extended Data Fig. 9a). In the Mock$_{VACC}$ group, the mean increase in body temperature between 1 and 4 d.p.i. (2 °C) was higher compared with that of the Anhui$_{VACC}$ (1.3 °C), AC-Anhui$_{VACC}$ (1.1 °C) and Giza$_{VACC}$ (0.9 °C) groups (Fig. 4c,d and Extended Data Fig. 10). Infectious virus was recovered from nose and throat swabs of ferrets from all groups, corroborating that intramuscular immunization does not prevent upper-respiratory-tract infection (Extended Data Fig. 9c,e). While infectious virus was isolated from respiratory tissues of all Mock$_{VACC}$ animals at 4 d.p.i. (Fig. 5a), significantly lower virus levels were observed in the trachea, bronchi and lungs of vaccinated ferrets (Fig. 5a). In fact, many vaccinated animals showed no infectious virus in trachea, bronchus and lung samples (Fig. 5a), indicating robust protection of vaccination against lower-respiratory-tract infection with the H5N1$_{Giza}$ virus, despite undetectable HI titres in some animals. The mean relative lung weight, reflective of pulmonary inflammatory infiltrate and oedema, was higher in the Mock$_{VACC}$ group compared with the vaccinated groups (Extended Data Fig. 9g). Histopathological lesions and lymphocyte presence in respiratory tissues were consistent with

viral replication and correlated with antigen expression detected by immunohistochemistry (IHC), as detailed in Supplementary Note 4, Supplementary Table 10 and Extended Data Fig. 11. Overall, vaccination strongly reduced disease severity and viral replication in the lower respiratory tract after H5N1$_{Giza}$ virus inoculation, irrespective of the vaccine used. However, the animals in the AC-Anhui$_{VACC}$ and Giza$_{VACC}$ groups showed significantly reduced mean virus titres in both the cerebrum and cerebellum compared with those in the Mock$_{VACC}$ group, while Anhui$_{VACC}$ animals did not. These results showcase the overall non-inferiority of the antigenically central antigen to the homologous Giza$_{VACC}$ and demonstrate its ability to prevent extrarespiratory spread of the H5N1$_{Giza}$ virus to the brain.

The H5N6$_{Sichuan}$ virus challenge resulted in more severe disease compared with the H5N1$_{Giza}$ virus. Ferrets in the Mock$_{VACC}$ group displayed reduced activity from 3 d.p.i. onwards (Extended Data Fig. 9b). At 4 d.p.i., two ferrets were found dead, and the remaining four exhibited difficulty walking and a large reduction in activity (Extended Data Fig. 9b). In the vaccinated groups, no significant changes in activity level score (Extended Data Fig. 9b) or clinical signs were observed after H5N6$_{Sichuan}$ virus inoculation. Moreover, the mean reduction in body weight experienced by AC-Anhui$_{VACC}$ (2%) and Sichuan$_{VACC}$ (1.5%) ferrets was significantly lower compared with that of Mock$_{VACC}$ ferrets (15.2%), while that of Anhui$_{VACC}$ ferrets (6.6%) was not (Fig. 4b). Moreover, Mock$_{VACC}$ ferrets exhibited a significantly higher temperature increase (2.2 °C) compared with AC-Anhui$_{VACC}$ (0.7 °C) and Sichuan$_{VACC}$ (0.5 °C) ferrets, but not compared with Anhui$_{VACC}$ ferrets (1.7 °C) (Fig. 4e,f and Extended Data Fig. 10). As expected, infectious virus was isolated from nose and throat swabs from 2 d.p.i. onwards in most animals across all

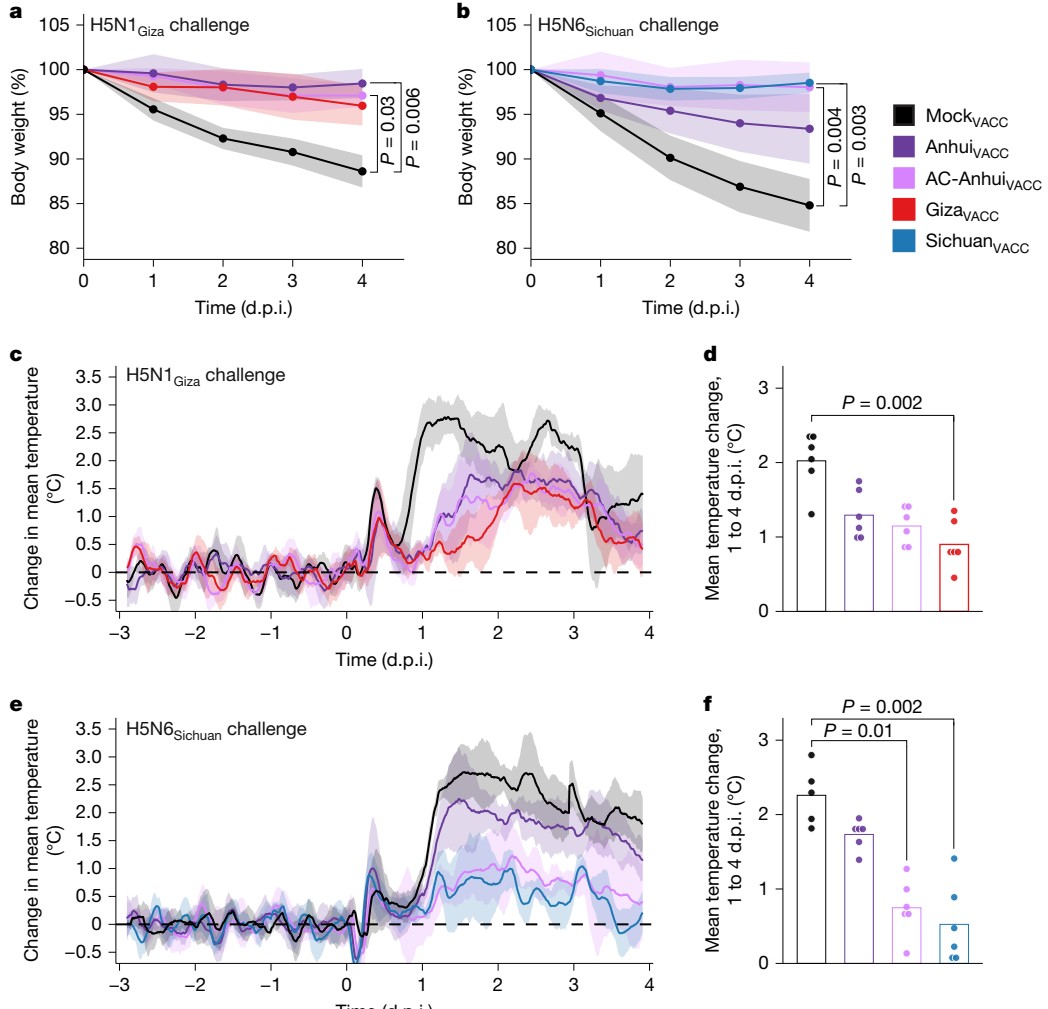

**Fig. 4 | Ferret body weight and temperature changes reveal non-inferior protection of AC-Anhui$_{VACC}$ vaccination compared with homologous vaccination. a,b,** Body weight expressed as a percentage of the starting body weight of ferrets ($n = 6$ per group) challenged with the H5N1$_{Giza}$ (**a**) or H5N6$_{Sichuan}$ (**b**) virus. The lines connect the daily arithmetic means. The shaded areas indicate the s.d. of the mean per group. **c,e,** Body temperature change from the baseline (mean temperature recorded during the 3 days before inoculation, indicated as a dashed line) of ferrets challenged with the H5N1$_{Giza}$ (**c**) or H5N6$_{Sichuan}$ (**e**) virus. Per group, the mean of individual 4-h sliding means is displayed. The shaded areas indicate the s.d. of the mean per group. **d,f,** The mean body temperature change from the baseline between 1 and 4 d.p.i. of

ferrets challenged with the H5N1$_{Giza}$ (**d**) or H5N6$_{Sichuan}$ (**f**) virus. The dots show data of individual animals. In **c**–**f**, $n = 6$ per group, except for Mock$_{VACC}$ with H5N6$_{Sichuan}$ challenge, for which $n = 5$ because of temperature probe malfunction. Data of the two deceased animals in the Mock$_{VACC}$ group were included in the visualization and analysis up until 3 d.p.i. Significant differences between groups were assessed using a Kruskal–Wallis test followed by a pairwise two-sided Dunn's test with Bonferroni correction for multiple comparisons. All pairwise comparisons were tested. For **a**, **b**, **d** and **f**, only statistically significant differences ($P < 0.05$) are indicated with the corresponding $P$ value. For **a** and **b**, areas under the curve were used for statistics.

groups, as observed for the H5N1$_{Giza}$ challenge (Extended Data Fig. 9d,f). In the Mock$_{VACC}$ group, infectious virus was isolated from respiratory tissues, liver and spleen of all animals, demonstrating extrarespiratory spread of the virus, confirmed by histopathological and IHC analysis (Fig. 5b, Extended Data Fig. 11, Supplementary Note 4 and Supplementary Table 10). Respiratory tract virus titres were generally lower in AC-Anhui$_{VACC}$ and Sichuan$_{VACC}$ ferrets as compared with Anhui$_{VACC}$ animals, with significant differences observed in lung and bronchus titres between Mock$_{VACC}$ and both Sichuan$_{VACC}$ and AC-Anhui$_{VACC}$ animals, but not between Mock$_{VACC}$ and Anhui$_{VACC}$ animals (Fig. 5b). The mean relative lung weight (Extended Data Fig. 9h) and histopathological and IHC analysis of lung tissue samples (Supplementary Note 4, Supplementary Table 10 and Extended Data Fig. 11) corroborated these results. AC-Anhui$_{VACC}$ and Sichuan$_{VACC}$ also protected all animals against extra-respiratory virus spread, while infectious virus was isolated from the liver and spleen of one Anhui$_{VACC}$ animal (Fig. 5b). Analogously to

the H5N1$_{Giza}$ virus challenge, AC-Anhui$_{VACC}$ conferred non-inferior protection as compared with Sichuan$_{VACC}$, despite more severe disease. Moreover, AC-Anhui$_{VACC}$ outperformed Anhui$_{VACC}$ in reducing infection severity, evidenced by reduced body weight loss, body temperature increase, virus titres, histopathological changes, antigen expression in the lungs and extrarespiratory virus spread.

## Discussion

Here we used antigenic cartography to visualize and quantify the antigenic evolution of A(H5) influenza viruses and design antigenically central vaccine antigens. In contrast to the directional evolution observed for human seasonal A(H3) influenza viruses[16], the A(H5) antigenic evolution exhibited non-directionality, requiring more than two dimensions to retrace complex evolution patterns, similar to other animal influenza viruses[24–27]. Discordance between antigenic and genetic evolution was

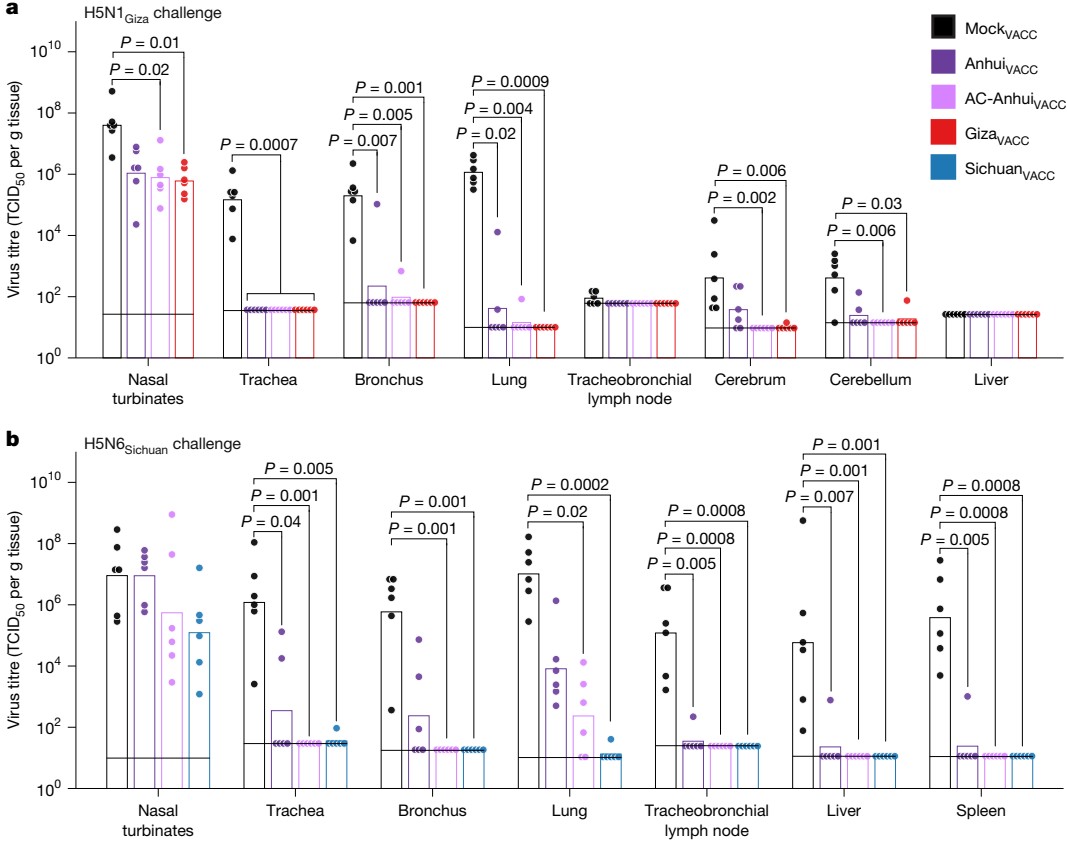

**Fig. 5 | Vaccination with AC-Anhui reduced infectious virus titres in ferret tissues equally well as homologous vaccination. a**, Vaccination–challenge study with the H5N1$_{Giza}$ virus. **b**, Vaccination–challenge study with the H5N6$_{Sichuan}$ virus. Data are colour coded on the basis of vaccine group as indicated in the legend. The bars represent the GMT (TCID$_{50}$ per g tissue) per group ($n = 6$). The dots represent titres in tissues of individual animals. The horizontal black lines indicate the detection limit for each tissue. Virus titrations were performed in four replicates. Significant differences between groups were assessed using a Kruskal–Wallis test followed by a pairwise two-sided Dunn's test with Bonferroni correction for multiple comparisons. All pairwise comparisons were tested. Only statistically significant differences ($P < 0.05$) are indicated with the corresponding $P$ value.

observed, reminiscent of findings on avian A(H7) influenza viruses[24] and other animal influenza viruses[27]. This suggests that A(H5) HA global antigenic evolution might be driven by only a few amino acid changes, as already proposed[25,28,29], and similar to human seasonal influenza viruses[30–32] and other animal influenza A viruses[24,27].

The evolutionary patterns of animal influenza viruses are difficult to understand and predict because the underlying drivers remain unclear. In contrast to in humans, in which population immunity buildup leads to the selection of drifted viruses evolving away from previously circulating variants, herd immunity is not expected to accumulate as such in poultry, allowing the co-circulation of antigenically diverse viruses. This is probably primarily attributable to the low reinfection likelihood given segregation of animal populations in time and location, shorter life spans, virus lethality and rapid renewal of susceptible populations. These aspects are particularly relevant for poultry, the primary hosts of A(H5) GsGd viruses. Recent enzootic establishment of A(H5) GsGd in various wild bird species[14] may result in different patterns of antigenic evolution. On the other hand, vaccination of poultry, practiced in some countries, may contribute to antigenic evolution of avian influenza viruses[33]. However, it is improbable that vaccination alone can fully explain the observed A(H5) GsGd viruses antigenic evolution, especially during the early 2004–2005 diversification period, when vaccination was in early implementation stages. Alternatively, antigenic evolution of A(H5) GsGd viruses could be partially a bystander effect of other adaptive processes, such as virus adaptation to systemically replicate or to host species switching. Understanding whether the other influenza virus antigen, NA, undergoes antigenic evolution in avian hosts would

be valuable to further identify the overall underlying drivers of avian influenza viruses evolution.

The diverse antigenic landscape of A(H5) influenza viruses poses considerable challenges to pandemic preparedness and vaccine design. Subtype-wide vaccines offering protection against drifted variants have been identified as crucial initial steps towards the development of truly universal vaccines[34]. Previous subtype-wide approaches involving the design of reconstructed ancestral HAs[35], genetic HA consensus antigens[36–41], mosaic HAs with conserved T and B cell epitopes[42] or phylogenetically central antigens[43] aimed to recapitulate A(H5) genetic diversity. These synthetic antigens induced broader antibody responses compared with wild-type comparators. However, they are representative of a population of genetic sequences rather than a population of antigenic properties. Given that the influenza virus antigenic phenotype is mainly governed by a few amino acid changes[25,31,32], genetic information at the level of the full HA may not be predictive. Here we focused on designing antigens based on antigenic phenotype rather than genetic information. Our observations indicated a relatively stable A(H5) antigenic space, which, along with a non-directional antigenic evolution, supported the design of antigenically central HA antigens. Toward this goal, we simultaneously altered the receptor binding specificity and glycosylation of non-central A(H5) HAs, inspired by natural viruses located centrally in antigenic space. Anticipating the emergence of an influenza pandemic virus with an α2,6-linked sialic acid binding specificity, we adjusted the receptor binding specificity of avian influenza vaccines to mirror that of human influenza viruses. The breadth and height of antibody responses after vaccination was

assessed against over a hundred A(H5) antigens, using the antigenic map as unique tool to visualize antibody response breadth. This enabled a comprehensive assessment of cross-reactivity, overcoming limitations posed by smaller datasets used in previous studies[35,36,41–44]. The CVAs demonstrated increased reactivity in HI assays and induced robust and central antibody responses. Vaccination with AC-Anhui$_{VACC}$ conferred non-inferior protection against two viruses genetically and antigenically distinct from homologous antigens. While some level of protection was observed in ferrets from the Anhui$_{VACC}$ groups, most probably due to non-neutralizing antibodies, the AC-Anhui$_{VACC}$ outperformed its wild-type counterpart in reducing H5N6$_{Sichuan}$ disease severity and H5N1$_{Giza}$ spread to the brain. In a recent study aimed at identifying the molecular determinants of antigenic differences between recent A(H5) Chinese poultry vaccine antigens, the authors identified a mutant A(H5) positioned in the middle of a Chinese clade 2.3.4.4 antigenic map, yet only one antigenic unit away from its wild-type counterpart[45]. Vaccination of chickens with this mutant HA offered protection against various clade 2.3.4.4 Chinese viruses, further reinforcing the idea that antigenically centrally located antigens may offer broader protection.

There are routes to further improve the immunogenicity and breadth of A(H5) antigenically central influenza vaccines. While AC-Anhui$_{VACC}$ provided good coverage within clade 2.3.4.4 antigenic space, coverage of clade 2.3.2.1 could be enhanced. Expanding antigenic space coverage could be achieved through heterologous prime–boost strategies using vaccines situated in different regions of the map[15], and/or by enhancing the immunogenicity and breadth of emerging variants using the same design as described here. Increased knowledge on the molecular determinants of antigenic change would also offer opportunities to further tune the positioning of vaccine antigens in space. Immunogenicity and breadth enhancement could be achieved by using different vaccine platforms. Here we chose to use inactivated vaccines, given that currently licensed A(H5) pre-pandemic vaccines are of this type. Whole-inactivated vaccines were more immunogenic than split-inactivated vaccines, as reported previously in unprimed individuals[46]. Antigenically central vaccine antigens could be used in conjunction with other vaccine platforms that enable de novo synthesis of viral proteins, which are necessary to elicit cellular immune responses and may induce higher B and T cell immune responses, such as mRNA vaccines[47,48] or vector-based vaccines[49]. While the only available data on immunogenicity of avian influenza mRNA vaccines in humans are underwhelming[50], there are currently insufficient data to fully evaluate the potential of this vaccine platform for avian influenza vaccines in humans. Efforts should be pursued to further develop platform alternatives to enhance activation of the cellular arm of the immune system. Alternative routes of administrations that may elicit mucosal immunity in addition to systemic immunity will also be important in designing vaccines reducing not only severe disease but also transmission. Lastly, our study has focused on the design of immunogenic and broad HAs. However, there is a growing body of evidence supporting the role of immunity against NA in conferring protection against influenza viruses[51–53]. Here we deliberately mismatched the vaccine NAs with the challenge virus NAs to specifically study the impact of our HA design on vaccine immunogenicity, cross-reactivity and protective capacity. However, ensuring matching of (pre-)pandemic vaccine NAs with that of emerging A(H5) viruses will be of the utmost importance. Recent human infections were caused by GsGd viruses carrying N1 or N6 NAs. While cross-reactive NA antibodies to avian N1 NAs have been detected in humans due to past infections and/or vaccinations with seasonal influenza A(H1N1) viruses[54,55], humans are most likely immunologically naive to N6 NAs. Understanding the immunogenicity and potential antigenic evolution of avian NAs could lead to the design of antigenically central NAs, which might be instrumental for developing more effective (pre-)pandemic vaccines.

We acknowledge two limitations. First, immunogenicity and cross-reactivity were solely evaluated by HI assays. Non-neutralizing antibody responses and T cell responses merit further study. Moreover, understanding monoclonal antibody reactivity profiles would provide deeper insights into mechanisms underlying the observed increased immunogenicity and breadth. Second, the relative contributions of altered receptor binding properties and glycosylation to increased immunogenicity remain unclear. Previous studies have reported increased immunogenicity attributed to α2,6-linked sialic acid binding, sometimes in synergy with glycosylation site removal[17,56–59]. These observations were made primarily with live attenuated vaccines, but mechanisms may differ in inactivated formulations, possibility involving enhanced binding to antigen-presenting cells.

The antigenic map presented here will offer possibilities to monitor the emergence of new A(H5) antigenic variants in the context of historical diversification. Assessing breadth of pre-pandemic vaccines is crucial given the diverse antigenic landscape of A(H5) viruses. Human clinical studies of A(H5) vaccines have generally assessed immune responses against a handful of outdated heterologous viruses[15]. The present antigenic map can inform more representative panels to assess vaccine coverage. Given the intrinsic low immunogenicity of natural avian influenza viruses, rationally designed immunogenic and broad A(H5) vaccines may offer better antibody responses compared with strain-matched wild-type vaccines. Antigenically central vaccines could serve as a prime in pre-pandemic time or be deployed early in a pandemic until a matched vaccine is available. In naive individuals, inducing primary responses central in antigenic space will confer variant-agnostic pre-immunity compared with vaccinating with a clade-specific vaccine. This is particularly important as primary responses will be boosted after heterologous exposure due to immune imprinting. In summary, we provide a proof of concept for designing and evaluating antigenically central A(H5) vaccines using antigenic cartography, a strategy that could be expanded to other zoonotic influenza viruses. The proof of concept presented here warrants follow-up with immunogenicity and breadth studies in humans to improve pandemic preparedness.

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

## Methods

### Phylogenetic tree construction

All available A(H5) HA nucleotide sequences and accompanying metadata were downloaded from the GISAID Data Science Initiative[60] and the Bacterial and Viral Bioinformatics Resource Center (BV-BRC)[61] databases on 5 May 2023. The HA sequences of three antigens present in-house (A/Vietnam/3218/2004, A/duck/Hong-Kong/1091/2011 and A/eurasian-wigeon/Netherlands/EMC-3/2014), which were not yet available through the above mentioned databases at the time, were added to the dataset manually and deposited to GISAID in hindsight. The dataset was then preprocessed using Pépinière (a Python (v.3.10.14) jupyter notebook available at GitHub (https://github.com/epiv-lab/pepiniere) and Zenodo[62]). This included deduplication of sequences present in both datasets based on identical accession numbers, identification and extraction of the open reading frame (ORF) corresponding to the longest ORF, and removal of sequences (1) without metadata or (2) shorter than 90% of the mean ORF length. Identical sequences were then grouped, and only the earliest (by isolation date) representative was retained. After preprocessing, sequences were aligned using MAFFT (v.7.515)[63], and the alignment was trimmed to the start and stop codons of the majority of sequences. Trimmed sequences were again filtered to remove identical sequences (retaining the earliest). The resulting dataset contained 14,896 sequences (Supplementary Table 1) and was realigned using MAFFT v.7.515 and maximum-likelihood trees were generated using IQ-Tree2 (v.2.1.4_beta)[64] with the GTR + F + R10 model (chosen by ModelFinder[65], implemented in IQ-Tree2) and 10,000 UFboot bootstrap approximations[66]. Trees were midpoint-rooted, annotated, and visualized using iTOL[67]. The display item was generated using ggtree (v.1.4.11)[68] in R (v.4.4.3). Genetic clades were predicted using LABEL[69] with the H5v2023 pre-release 1 (2023-05-05) module, courtesy of S. Shepard, and this prediction was used to annotate the tree.

### Cells

Cells were maintained as described previously[24]. 293T cells (ATCC) were cultured in Dulbecco's modified Eagle's medium (DMEM) (Lonza) supplemented with 10% FCS (Sigma-Aldrich), 1× non-essential amino acids (Lonza), 1 mM sodium pyruvate (Gibco), 2 mM L-glutamine (Lonza), 100 U ml$^{-1}$ penicillin, 100 µg ml$^{-1}$ streptomycin (Lonza) and 0.5 mg ml$^{-1}$ geneticin (Invitrogen). Madin–Darby canine kidney (MDCK) cells (ATCC) were cultured in Eagle's minimal essential medium (EMEM) (Lonza), supplemented with 10% FCS, 1× non-essential amino acids (Lonza), 1.5 mg ml$^{-1}$ sodium bicarbonate (Lonza), 10 mM HEPES (Lonza), 2 mM L-glutamine (Lonza), 100 U ml$^{-1}$ penicillin and 100 µg ml$^{-1}$ streptomycin (Lonza). Cells were cultured at 37 °C, 5% CO$_2$, and passaged twice weekly. Cells were not authenticated. All cell banks were tested negative for mycoplasma.

### Generation of plasmids

To generate plasmids for recombinant virus production, viral RNA was isolated from in-house available virus isolate stocks using the High Pure RNA Isolation Kit (Roche) according to the manufacturer's instructions. The RNA was then used to generate viral copy DNA (cDNA) using SuperScript IV reverse transcriptase (Thermo Fisher Scientific) according to the manufacturer's instructions; from the cDNA, individual HA or NA gene segments were amplified using segment-specific PCR (polymerase chain reaction) primers[70] and the PfuUltra II Fusion HS DNA Polymerase (Agilent) according to the manufacturer's instructions. Individual viral gene segments were cloned into a previously described modified pHW2000 plasmid[71] by restriction-site-based cloning or seamless cloning using the GeneArt Seamless Cloning kit (Thermo Fisher Scientific). If the respective virus isolate was not present in-house, synthetic genes containing HA sequences with a monobasic cleavage site were synthesized by BaseClear or Integrated DNA Technologies. When applicable,

specific mutations were introduced in the HA genes and /or the HA multibasic cleavage site (MBCS) was removed by site-directed mutagenesis using the PfuUltra II Fusion HS DNA Polymerase (Agilent) and specific primers. Throughout the Article, A(H5) numbering is used to refer to specific amino acid positions[72]. To produce reverse genetics plasmids for A(H5N6) A/Sichuan/26221/2014 (Sichuan, accession number: EPI_ISL_163493), all eight gene segments were amplified from cDNA by PCR using specific primers[70] and cloned into our bidirectional reverse genetics plasmid. To produce reverse genetics plasmids for A(H5N1) A/duck/Giza/15292S/2015 (Giza, accession number: EPI_ISL_257168), viral RNA was extracted, and all eight gene segments were amplified from cDNA by PCR using specific primers[70] and cloned into our bidirectional reverse genetics plasmid. Non-coding regions, which are not part of the abovementioned sequences, were sequenced after RNA circularization as described previously[73]. The non-coding regions used are listed in Supplementary Table 11.

### Benefit sharing of synthetic constructs and viruses

Before the start of this work, we discussed and publicly announced our plans to generate synthetic HA constructs and produce recombinant A(H5) viruses and A(H5)-specific ferret sera through the GISAID website (https://gisaid.org/collaborations/collaboration-on-h5-antigenic-cartography/). Specifically, we made the commitment that the synthetic HA constructs, reverse genetics viruses and ferret sera will be shared with the laboratories that contributed the genome sequence data to GISAID[60]. We committed to publish the antigenic maps with open access to the public. We also indicated that reagents may be provided to other researchers, including National Influenza Centers and global reference laboratories, after assurance that the originating laboratory, where the clinical specimen or virus isolate was first obtained, and the submitting laboratory, where sequence data have been generated and submitted through the GISAID Data Science Initiative, are fully recognized, to ensure fair attribution of contributions to the results benefitting from the data. We are indebted to GISAID and all scientists contributing to the GISAID Data Science Initiative, without whom this work would not have been possible. We thank the governments and scientists of Austria, Bangladesh, Cambodia, China, Egypt, Germany, Ghana, Indonesia, India, Iraq, Japan, Nepal, Nigeria, Mongolia, Russia, Scotland, South Africa, Sweden, Turkey, United States and Vietnam for their contributions that made this research possible.

### Biosafety

All experiments were reviewed by the Erasmus MC Institution Review Entity (IRE), in accordance with the US Government September 2014 Dual-Use of Research of Concern (DURC) policy. The Erasmus MC IRE concluded that the studies described here were not falling under any of the seven DURC categories. Recombinant viruses that contained the HA (without MBCS) and/or NA of interest in the background of the attenuated vaccine strain A/Puerto Rico/8/1934 (PR/8) or the high-yield version thereof (PR/8 HY, 76) (Supplementary Table 2) were handled under biosafety level 2 (BSL2) conditions in agreement with national regulations. HPAIV wild-type isolates used in HI assays (Supplementary Table 2) were handled under BSL3 conditions. For the ferret challenge experiments, H5N6$_{Sichuan}$ and H5N1$_{Giza}$ recombinant viruses, which contained all eight wild-type gene segments of a single virus, were produced. These experiments were performed in the enhanced animal biosafety level 3 (ABSL3+) facility of the Erasmus University Medical Center as described previously[74].

### Recombinant virus production

Recombinant influenza viruses were generated by reverse genetics using eight bidirectional plasmids as described previously[24,71]. One day before transfection, about $3 \times 10^6$ 293T cells were seeded in gelatin-coated 10 cm culture dishes. Calcium-phosphate-mediated transfection was used to deliver a total of 40 µg of plasmid DNA per

dish. About 16 h after transfection, the cells were washed once with PBS and fresh medium containing 2% FCS with 200–350 µg ml⁻¹ *N*-tosyl-ʟ-phenylalanine chloromethyl ketone (TPCK)-treated trypsin (Sigma-Aldrich) was added. Virus stocks were generated by inoculating either MDCK cells or 11-day-old embryonated chicken eggs with dilutions of the supernatant collected from the 293T cells 3 days after transfection or virus isolates. Virus stock production in MDCK cells was performed using EMEM medium containing the same supplements as described above, but without FCS and with the addition of 20–35 µg ml⁻¹ TPCK-treated trypsin, referred to as infection medium. MDCK supernatants or embryonated egg allantoic fluids were collected 2–3 d.p.i. and centrifuged at 2,100g for 10 min to remove cellular debris. The presence of virus was confirmed by haemagglutination assays using 1% turkey red blood cells (TRBCs, from in-house turkeys) in PBS. Sequences from all plasmids and from the non-PR/8 and PR/8 HY genes, that is, HA and NA of interest, of all virus stocks were confirmed with Sanger sequencing using the BigDye Terminator v.3.1 Cycle Sequencing Kit (Applied Biosystems) and the 3500xL Genetic Analyzer (Applied Biosystems).

## Virus titrations

Virus titrations were performed in MDCK cells as described previously[74]. In brief, flat-bottom 96-well plates containing confluent MDCK cells were inoculated with tenfold serial dilutions of the samples and incubated for 1 h at 37 °C under 5% $CO_2$. Cells were washed once with PBS and 200 µl of infection medium was added to each well. After 3 days of incubation at 37 °C, 5% $CO_2$, the presence of virus in the supernatants was determined using HA assays to determine the $TCID_{50}$. Virus titrations of the virus stocks were performed in ten replicates, and those of respiratory swabs and tissue homogenates from the vaccination-challenge experiments in four replicates. Virus titres were read out blindly.

## Vaccine production

Vaccines were produced with recombinant viruses containing a mutated or wild-type HA, without MBCS, in the PR/8 HY background. For the initial screen of CVAs, whole-inactivated vaccines were generated with the corresponding matched NA. For the vaccination-challenge experiment, split-inactivated vaccines were used. To isolate the effect of varying the HA antigen in these studies, the NA present in the split-inactivated vaccines was mismatched with that of the challenge virus. Specifically, the vaccines in the $H5N1_{Giza}$ challenge contained the N6 NA of $H5N6_{Sichuan}$, and the vaccines in the $H5N6_{Sichuan}$ challenge contained the N1 NA of $H5N1_{Giza}$.

Whole-inactivated and split-inactivated vaccines were generated as described previously[52]. Eleven-day-old embryonated chicken eggs were inoculated with the virus of interest. Allantoic fluid was collected 2 d.p.i. and centrifuged for 10 min at 2,100g to remove cellular debris. Subsequent centrifugation steps were performed at 124,000g (SW 32 Ti, Beckman Coulter) at 4 °C, unless indicated otherwise. The allantoic fluid was concentrated on a 60% sucrose cushion by centrifuging for 2 h. Subsequently, resuspended sucrose cushions from multiple tubes were pooled and loaded onto 60/50/40/30/20% sucrose gradients, and centrifuged overnight at the lowest deceleration setting. The virus band, located on top of the 30% sucrose layer, was collected, diluted in PBS and subsequently pelleted by centrifugation for 2 h to remove the sucrose. The pellet was dissolved in either PBS (whole-inactivated vaccines) or PBS with 2% *N*-decanoyl-*N*-methylglucamine (Mega10, Sigma-Aldrich) (split-inactivated vaccines) to solubilize the viral membrane proteins. Incubation with PBS with 2% Mega10 was performed for 1 h at 37 °C. For both whole- and split-inactivated vaccines, the dissolved pellets were transferred to dialysis chambers (Slide-A-Lyzer Dialysis Cassettes, 10 K MWCO, Thermo Fisher Scientific) and subsequently submerged in PBS containing 0.01% formalin for 3 days. Subsequently, the dialysis chambers were immersed in PBS for a day, during which the PBS was refreshed twice. The resulting vaccines were aliquoted

and stored at −80 °C. Vaccine inactivation was confirmed by two serial blind passages on MDCK cells and/or in embryonated chicken eggs.

Total protein content was determined using the Pierce bicinchoninic acid total protein analysis kit (Thermo Fisher Scientific). For the whole-inactivated vaccines, the absolute and relative HA content was estimated from SDS–PAGE protein gels using a BSA standard and stained with Instant Blue (Expedeon). The absolute HA content of the split-inactivated vaccines was estimated from SDS–PAGE using a BSA standard and stained with Instant Blue (Expedeon). The relative HA content of split-inactivated vaccines was determined with mass spectrometry, using a protocol based on a previous study[75] with modifications as described previously[52]. Diluted vaccines (10 µl of 125 µg ml⁻¹ of total protein) were mixed 1:1 with 0.2% RapiGest (Waters) and denatured for 5 min at 100 °C. After cooling to room temperature, 5 µl of sequence-grade modified trypsin solution (0.4 µg µl⁻¹; Promega) was added, and samples were incubated at 37 °C for 2 h. Digests were allowed to cool, and 55 µl of 0.5% trifluoroacetic acid was added. The samples were subsequently analysed by a nano-liquid chromatography (nano-LC) Ultimate 3000 system (Thermo Fisher Scientific) coupled to the Orbitrap Fusion Lumos mass spectrometer (Thermo Fisher Scientific).

Data from initial screens were used to select three peptides for stable isotope (SI) labelling (LVLATGLR, VNSIIDK and TLDFHDSNVK), based on intensity, length, sequence and sequence conservation within the A(H5) HA subtype. Digested vaccines were spiked with SI-labelled peptides with heavy lysine or arginine (Pepscan) at a final concentration of 50 fmol µl⁻¹, and measured on the nano-LC system (Ultimate 3000; Thermo Fisher Scientific) combined with an Orbitrap Fusion Eclipse Tribrid mass spectrometer (Thermo Fisher Scientific). For each peptide, the ratio between the SI-labelled peptide and the endogenous peptide was calculated, and was subsequently used to determine the concentrations of the endogenous peptide in the vaccines. The HA concentration based on the three individual peptides was averaged for each vaccine, and subsequently used to determine the relative HA content of the vaccines.

## Ferret experiments

Ferret (*Mustela putorius furo*) experiments were performed in strict compliance with the Dutch legislation on the protection of animals used for scientific purposes (2014, European Union directive 2010/63/EU implemented). Experiments were performed at the Erasmus Medical Center in Rotterdam, the Netherlands under a project license accredited by the Dutch competent authority (license number AVD101002015340). Study protocols were approved by the Erasmus Medical Center Animal Welfare Body (permit numbers 15-340-01, -04, -06, -22, -23 and -24). Ferrets were seronegative for Aleutian disease, seasonal influenza A(H1N1), A(H3N2) and B viruses. For all ferret experiments, animals were randomly allocated to the different groups. Ferret experiments were not performed blindly because regulations required knowledge of the animal treatment for biosafety reasons.

**Serum production.** Ferret sera were generated as described previously[24,25] in class III isolators under BLS3 conditions using recombinant viruses unless indicated otherwise (Supplementary Tables 2 and 3). Recombinant viruses were produced carrying the HA (without MBCS) and the closest matching NA present in-house, in the background of PR/8 or PR/8 HY. In brief, male ferrets (at least six months old) were inoculated intranasally by applying dropwise 250 µl of virus stock per nostril. Unless indicated otherwise (Supplementary Table 2), a boost was administered after 14 days, by subcutaneously injecting a total of 250 µl concentrated virus combined with 250 µl TiterMax Gold adjuvant (Sigma-Aldrich) at two different spots in the back.

The concentrated virus used for the subcutaneous boost was prepared by inoculating five 11-day-old embryonated chicken eggs per virus. The allantoic fluid was collected 2 d.p.i. and cleared from debris

by centrifuging for 10 min at 2,100$g$. About 36 ml of the cleared allantoic fluid was concentrated by centrifuging for 2 h at 124,000$g$ (SW 32 Ti, Beckman Coulter), and the resulting pellet was resuspended in 700 µl PBS. Ferrets were terminally bled through cardiac puncture 14 days after the subcutaneous boost, or 14 days after the intranasal inoculation if no boost injection was performed. Before virus inoculation, subcutaneous boost injection and the terminal bleed, ferrets were anaesthetized with ketamine and medetomidine (10 and 0.05 mg per kg body weight, respectively), the latter which was antagonized with atipamezole (0.25 mg per kg body weight).

The blood samples were collected in VACUETTE CAT Serum Separator Clot Activator tubes (Greiner Bio-One), incubated at least for 15 min to allow clotting, and centrifuged for 15 min at 2,000$g$ to obtain the serum.

**Vaccination–challenge studies.** Vaccination experiments were performed similarly as described previously[52]. Six- to 12-month-old female ferrets were used for vaccination studies ($n$ = 2 for vaccination-only experiments and $n$ = 6 for vaccination–challenge experiments). Ferrets were vaccinated twice intramuscularly on day 0 (prime) and day 28 (boost) with 250 µl of whole or split-inactivated vaccine estimated to contain about 7.5 µg HA, adjuvanted with 250 µl AddaVax (Invivo-Gen), which was equally divided between the two hind legs. For the mock-vaccinated groups, animals were vaccinated with 250 µl of PBS adjuvanted with 250 µl AddaVax. Before each vaccination, a blood sample was obtained through the cranial vena cava and serum was obtained as described above (pre- and pre-boost sera). The pre-sera were tested in HI assays as described below against seasonal influenza A(H1N1), A(H3N2) and B viruses (using vaccine strains of the respective year), as well as PR/8 recombinant viruses carrying A(H5) HAs from the respective vaccines and challenge virus, if applicable. Pre-sera were negative in HI assays against the tested viruses. The pre-boost sera of the ferrets from the challenge experiments were titrated in HI assays against PR/8 recombinant viruses with three vaccine antigens employed in the respective study.

For vaccination-only experiments, ferrets were euthanized 28 days after the boost vaccination through a cardiac puncture, and post-boost sera were obtained from the whole blood as described above.

For vaccination–challenge experiments, DST micro-T temperature loggers (Star-Oddi) were surgically implanted into the abdominal cavity of the ferrets 14 days after the prime vaccination. Serum samples were collected from whole blood sampled through the cranial vena cava 1 week before inoculation (post-boost sera) (Extended Data Fig. 8 and Supplementary Tables 6 and 7). Vaccinated ferrets were then transferred to class III isolators for acclimatization a week before inoculation with the challenge virus. Ferrets were inoculated intranasally and intratracheally with wild-type recombinant viruses containing all eight segments of the respective challenge virus. The inoculation doses were $10^{5.5}$ and $10^{3.4}$ TCID$_{50}$ per animal for the H5N1$_{Giza}$ and H5N6$_{Sichuan}$ virus, respectively, divided over 3 ml intratracheally and 250 µl in each nostril. These doses were determined before the challenge in a pilot experiment using three ferrets per tested dose. The doses were selected to induce a reproducible and consistent infection of the upper and lower respiratory tracts. Subsequently, daily nose and throat swabs were collected under light ketamine anaesthesia, and body weight and activity level score were monitored daily as described previously[74,76]. Body temperature was recorded every 10 min by the implanted temperature loggers. Then, at 4 d.p.i., ferrets were euthanized through cardiac puncture, after which tissues (selected based on virus detection in the pilot studies) were collected for virological and pathological analysis as described previously[74]. For virological analysis, the right nasal turbinates, trachea, right bronchus, right lung lobes, tracheo-bronchial lymph node and liver (for both challenges), right cerebrum and right cerebellum (for the H5N1$_{Giza}$ challenge only), and the spleen (for the H5N6$_{Sichuan}$ challenge only) were collected. For pathological examination, the left nasal turbinates, trachea, left bronchus and left lung lobes were collected.

During blood collection, vaccination, virus inoculation and cardiac puncture, ferrets were anaesthetized with a mixture of ketamine and medetomidine, and antagonized with atipamezole, as described above.

### Histopathology and IHC

After necropsy, tissues were stored in 10% neutral-buffered formalin (lungs after careful inflation with formalin) for at least 2 weeks, after which the tissues were embedded in paraffin. Slides (4 µm) were made, and subsequent slides were either stained with haematoxylin and eosin or used for IHC as described previously[77]. In brief, after deparaffinization, antigen retrieval and blocking of endogenous proteases, slides were incubated for 1 h at room temperature with either a primary antibody against influenza A virus nucleoprotein (Hb65, American Type Culture Collection, H16-L10-4R5) or a mouse IgG2a isotype control (R&D, MAB003), diluted in PBS with 0.1% BSA (1:400 and 1:200, respectively). After three washes with PBS with 0.05% Tween-20, the slides were incubated for 1 h at room temperature with a goat anti-mouse IgG2a secondary antibody coupled to horseradish peroxidase (HRP) (Bio-Rad, Star133A), diluted 1:100 in PBS with 0.1% BSA. HRP was revealed using 3-amino-9-ethylcarbazole and a haematoxylin counterstain was performed. A lung section from an animal experimentally infected with 2009 pandemic A(H1N1) virus was used as positive control in each staining experiment.

The pathological changes and the presence of viral antigen in respiratory tissues were blindly scored in a semi-quantitative manner by a veterinary pathologist. Semi-quantitative assessment of influenza virus-associated inflammation in the lungs (four slides with longitudinal section and cross-section of cranial and caudal lobes per animal) was performed in a blinded manner on every slide as reported earlier[78]. The extent of alveolitis and alveolar damage was scored as follows: 0, 0%; 1, 1–25%; 2, 25–50%; 3, >50%. The severity of alveolitis, bronchiolitis, bronchitis, bronchial adenitis, tracheitis and rhinitis were scored as follows: 0, no inflammatory cells; 1, few inflammatory cells; 2, moderate numbers of inflammatory cells; 3, many inflammatory cells. The presence of alveolar oedema, alveolar haemorrhage and type II pneumocyte hyperplasia were scored as follows: 0, no; 1, yes. Finally, the extent of peribronchial, peribronchiolar and perivascular infiltrates were scored as follows: 0, none; 1, one to two cells thick; 2, three to ten cells thick; 3, more than ten cells thick. Semi-quantitative assessment of influenza virus antigen expression in the lungs was performed as reported earlier[79]. For the alveoli, 25 arbitrarily chosen fields of lung parenchyma of the 4 lung sections per animal were blindly examined by light microscopy, using a ×20 objective, for the presence of influenza virus nucleoprotein. The cumulative scores for each animal were presented as a percentage corresponding to the number of positive fields. The percentage of positive epithelium in the bronchi and bronchioles was estimated on all four lung slides and averaged per animal. The percentage of positively staining epithelium in the nose and trachea was estimated for one slide.

### Serological assays

HI assays were performed with recombinant viruses in PR/8 or PR/8 HY background and virus isolates (Supplementary Table 2) as described previously[24] using in-house TRBCs. Sera were treated overnight at 37 °C with five volumes of a *Vibrio cholerae* filtrate (generated in-house) containing receptor-destroying enzyme, to prevent aspecific inhibition. After inactivation for 1 h at 56 °C, sera were adsorbed using an equal volume of 10% TRBCs for 1 h at 4 °C to prevent aspecific agglutination. Twofold serial dilutions of sera in PBS were prepared in round-bottom 96-wells plates starting at 1:20 in a volume of 50 µl. To each well, 25 µl of virus, diluted in PBS to 4 hemagglutinating units, was added. After incubation for 30 min at 37 °C, 25 µl of 1% TRBCs was added to each well. Plates were subsequently incubated for 1 h at 4 °C before reading the HI titre. The HI titre was determined as the reciprocal value of the highest serum dilution which completely inhibited TRBC agglutination.

HI titres were read out blindly. For the calculation of GMTs, threshold titres of <10 were converted to 5 unless stated otherwise.

Virus neutralization assays were performed in MDCK cells as described previously[24,32]. First, sera were incubated for 30 min at 56 °C to inactivate complement. Twofold serial dilutions of sera in PBS, starting at 1:10, were combined with 100 $TCID_{50}$ of virus, and incubated for 2 h at 37 °C. Subsequently, the virus–serum mixtures were added to flat-bottom 96-wells plates containing confluent MDCK cells previously washed once with PBS. After incubation for 2 h at 37 °C and 5% $CO_2$, cells were washed once with PBS, and 200 µl infection medium per well was added. The plates were incubated at 37 °C under 5% $CO_2$, and the presence or absence of virus in the supernatants was determined after 3 days using HA assays with TRBCs. The virus neutralization titre was determined as the reciprocal value of the highest serum dilution for which no virus in the supernatants was detected. Virus neutralization titres were read out blindly. Virus neutralization assays were performed in duplicate, and the arithmetic means of $\log_2$ titres were calculated.

## Antigenic cartography and antibody profiles

Antigenic maps were constructed from HI data using a multidimensional scaling algorithm as described previously[16] using the R package Racmacs (v.1.2.3)[80]. First, HI titres are converted to a distance matrix (HI table distances) by (1) dividing each HI titre by 10 and applying a $\log_2$ transformation (hereafter, $\log_2$-transformed HI titres); and (2) subtracting each $\log_2$-transformed titre to the highest one for each serum. Secondly, multidimensional scaling algorithms are used to find the best set of map coordinates to represent the distances from the distance matrix most closely. For each optimization, antigens and sera, hereafter named points, are randomly placed in $n$-dimensional space, and coordinates are optimized from these starting conditions using the $L$-Broyden–Fletcher–Goldfarb–Shanno algorithm, minimizing the sum of the squared differences between HI table distances and the map distances (Euclidian distance between points in the $n$-dimensional space). In an antigenic map, every direction represents antigenic distance, and one antigenic unit corresponds to a twofold change in HI titre.

Unless stated otherwise, antigenic maps were computed using the 'make.acmap' function, with 1,000 optimization runs in three dimensions, and the minimum column basis set to zero. The antigenic map was validated using several tests, which are described in Supplementary Notes 2 and 3. Total map stress was extracted using the mapStress function and individual antigen stresses were extracted using the agStress function. HI table distances and pairwise antigen–serum Euclidian distances in the antigenic map were extracted using the tableDistances and mapDistances functions, respectively. When threshold HI titres (that is, <10) are converted to table distances in the process of making an antigenic map, the resulting values are not an exact distance but a 'greater-than' value, that is, thresholded distance. To include these points in the visualization in scatter plots of HI distances versus map distances (Extended Data Fig. 3a), and corresponding regression coefficient ($R^2$) calculations (Extended Data Figs. 1c and 3a), these values were converted to the thresholded distance increased by 1 on the $\log_2$ scale, for example, a threshold distance of >7 on the $\log_2$ scale is converted to an 8. To compute distances between points in the map, antigen and serum coordinates were extracted using the agCoords and srCoords functions, respectively. The dist function (base R) was subsequently used to compute pairwise Euclidean distances between points in the map. The map centre was determined by computing the mean $x$, $y$ and $z$ of the antigen coordinates. Pairwise genetic hamming distances, that is, the number of amino acid differences between two antigens, were computed using the 'stringDist' function (method = 'hamming') from the Biostrings package (v.2.74.1)[81].

To average and visualize the immune response of multiple ferrets belonging to the same experimental group (referred to as a mean serum), the GMTs of multiple vaccination sera against each individual antigen were computed to generate antibody profiles. Individual threshold titres were first converted to the closest possible numerical titre (for example, <10 to a 5) and, subsequently, GMTs were calculated. For the visualization of the reactivity of post-vaccination sera using the A(H5) antigenic map, antigenic maps were optimized with datasets containing the antigenic map HI data and HI data of a single post-vaccination serum or mean data of multiple post-vaccination sera as described above. The resulting maps including individual or mean post-vaccination sera data shared similar conformations, also corresponding to that of the antigenic map (the mean median Procrustes distance between each map with post-vaccination data and the antigenic map was 0.15 AU). Thus, to generate displays in which the positions of post-vaccination sera were visualized without changing the position of the antigens or sera in the antigenic map, maps, containing either individual or merged HI data, were superimposed onto the antigenic map using the mergeMaps function with the frozen-merge method. These maps were then used to visualize and analyse the reactivity of the post-vaccination sera to antigens in the antigenic map using custom R code (https://github.com/epiv-lab/H5-antigenic-evolution and Zenodo[82]). For analysis of mutant antigens, antigenic maps were computed with datasets containing HI data of a single mutant antigen in addition to the antigenic map dataset. The resulting optimized maps were used to calculate the distances described in the text. For visualization in Supplementary Data 5b, superimposed maps were generated as described above for the visualization of post-vaccination sera. The displays in Fig. 3 and Supplementary Data 8–10 were generated by superimposing an optimized map containing an individual single or mean post-vaccination serum, as described above, on the map from Supplementary Data 5b. Supplementary Video 1 was generated using the antigenic map displayed in Fig. 1b and Supplementary Data 2 using custom R code (https://github.com/epiv-lab/H5-antigenic-evolution and Zenodo[82]). In brief, individual frames were generated by taking screenshots of rotating r3js (v.0.0.2)[83] antigenic map displays using R package webshot2 (v.0.1.2)[84], which were subsequently assembled into a video using ffmpeg.

## Resialylated TRBC assay

Resialylated TRBC assays were performed as described previously[74]. The pellet of 1.25 ml of 1% TRBCs was resuspended in 62.5 µl PBS and incubated for 1 h at 37 °C with 50 µl of 1 mU µl$^{-1}$ *V. cholerae* neuraminidase (Roche) and 10 µl 0.1 M $CaCl_2$ to remove all sialic acids from the TRBCs. After two washes with PBS, the TRBCs were combined with 3.75 µl of 30 mM CMP-sialic acid (Merck), and either 5 µl of α2,3-sialyltransferase (recombinant human ST3GAL6, Fc Chimera, R&D systems) or 5 µl α2,6-sialyltransferase (Recombinant Human ST6GAL1 (amino acids 44–406) Protein, R&D Systems), and PBS up to 75 µl. Alternatively, 1 µg of in-house generated *Pasteurella multocida* sialyltransferase 1 (Pmst1) M144D (α2,3-sialyltransferase)[85] or Pmst1 M144L P34H (α2,6-sialyltransferase)[86] was used. TRBCs were incubated at 37 °C for 2 h with the commercial enzymes or for 4 h with the in-house generated enzymes. After resialylation, TRBCs were washed twice with PBS, and resuspended in PBS containing 1% BSA to a final concentration of 0.5% TRBCs. Besides α2,3- and α2,6-linked sialic-acid-specific TRBCs, untreated and VNCA-treated TRBCs were taken along as controls when assessing the binding preference of viruses in HA assays. Moreover, in each independent assay, a minimum of three control viruses was used: one with α2,3-linked sialic acid specificity, one with α2,6-linked sialic acid specificity, and one with dual binding specificity.

## Data visualization and statistics

Data were visualized with Racmacs (v.1.2.3)[80], r3js (v.0.0.2)[83] and/or ggplot2 (v.3.5.1)[87] in R v.4.4.3 (used throughout). The interactive figure files in the Supplementary Information (Supplementary Data 2–10) were generated with flexdashboard (v.0.6.2)[88] in R. Statistical analyses were performed using the base R functions for the Kruskal–Wallis test and linear regressions. For the comparison of multiple experimental

groups, the Kruskal–Wallis test was first performed. If positive, pairwise two-sided Dunn's tests with Bonferroni correction were then performed (R packages FSA (v.0.10.0)[89] and dunn.test (v.1.3.6))[90] to assess the significance of differences between two experimental groups.

## Reporting summary

Further information on research design is available in the Nature Portfolio Reporting Summary linked to this article.

## Data availability

All data supporting the findings of this study are available in the Article and its Supplementary Information. Sequence accession numbers are available in Supplementary Table 1. Supplementary Data 1–10 are also available online (https://epiv-lab.github.io/H5-antigenically-central-vaccine/). Source data are provided with this paper.

## Code availability

Custom code used for data analysis is available at GitHub (https://github.com/epiv-lab/H5-antigenic-evolution; https://github.com/epiv-lab/pepiniere) and Zenodo[62,82].

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

**Acknowledgements** We acknowledge all data contributors, including the authors and their originating laboratories responsible for obtaining the specimens, and their submitting laboratories for generating the genetic sequence and metadata and sharing through the GISAID Initiative, on which this research is based. In particular, we acknowledge P. Bogner for discussions and advice on appropriate benefit sharing. We thank staff at the following institutes for providing viruses and/or sequences for phenotypic characterization: St Jude Children's Hospital, the Southeast Poultry Research Laboratory, Center for Disease Control and Prevention, University of Wisconsin-Madison, Cambodian Pasteur Institute, Hong-Kong University, Queen Mary Hospital, Chinese Center of Disease Control and Prevention, Vaksindo Satwa Nusantara, Umeå University, National Public Health Laboratory Luxembourg, WHO CC for Reference and Research on Influenza, Crick Institute London, Friedrich-Loeffler Institut and the Austrian Agency for Health and Food Safety. We thank the staff at Hong-Kong University and University of Wisconsin-Madison for providing cDNA/viruses used for the challenge experiments; the members of the Erasmus Laboratory Animal Science Center staff for animal care, and specifically V. Vaes and D. Akkermans for their assistance in animal experiments; P. van Run and A. de Bruin for their help in processing the animal tissues for histology and IHC; D. van Mourik, R. Scheuer and M. Pronk for their technical assistance; and D. van de Vijver for input on statistical analyses; T. Luider and L. Dekker for performing the MS experiments; S. Shepard for sharing a pre-released version of LABEL for A(H5) clade classification; G. Neumann and Y. Kawaoka and their team for discussions and for providing the high-yield PR/8 backbone. Molecular graphics and analyses in the peer review file were performed using UCSF ChimeraX, developed by the Resource for Biocomputing, Visualization, and Informatics at the University of California, San Francisco, with support from National Institutes of Health R01-GM129325 and the Office of Cyber Infrastructure and Computational Biology, National Institute of Allergy and Infectious Diseases (https://www.rbvi.ucsf.edu/chimerax). This research was funded by the Biomedical Advanced Research and Authority (BARDA) contract number HHSO100201500033C; National Institute of Health, National Institute of Allergies and Infectious Diseases (NIH-NIAID) contract number HHSN272201400008C and 75N93021C00014; European Union's Horizon Europe FARM2FORK research and innovation program Kappa Flu grant number 101084171; and European Union's EU4Health program DURABLE grant number 101102733. S.L.J. was supported by the Medical Research Council Pre-doctoral Clinical Research Training Fellowship G105305.

**Author contributions** A.K., D.J.S., R.A.M.F. and M.R. conceptualized the project. A.K., S.H.W., S.T., S.L.J., D.F.B., M.F., D.J.P., J.M.A.v.d.B., S.H. and M.R. designed the experiments. A.K., T.M.B., S.v.d.V., M.I.S., W.F.R., D.d.M., M.E.R., P.L., J.M.A.v.d.B., S.H. and M.R. performed in vitro and in vivo experiments. A.K., S.T., S.L.J., D.F.B., M.F., D.J.P. and M.R. performed data analysis and interpretation. Final data visualization was done by A.K., S.H.W. and S.T. Custom code was written by S.H.W. and S.T.; M.R. managed the project. A.K. and M.R. wrote the initial manuscript. A.K., S.H.W., S.T., S.L.J., T.M.B., D.F.B., M.F., S.v.d.V., M.I.S., W.F.R., D.J.P., D.d.M., M.E.R., P.L., J.M.A.v.d.B., S.H., D.J.S., R.A.M.F. and M.R. reviewed and edited the manuscript. D.J.S., R.A.M.F. and M.R. acquired funding.

**Competing interests** The authors declare no competing interests

**Additional information**
**Correspondence and requests for materials** should be addressed to Mathilde Richard.

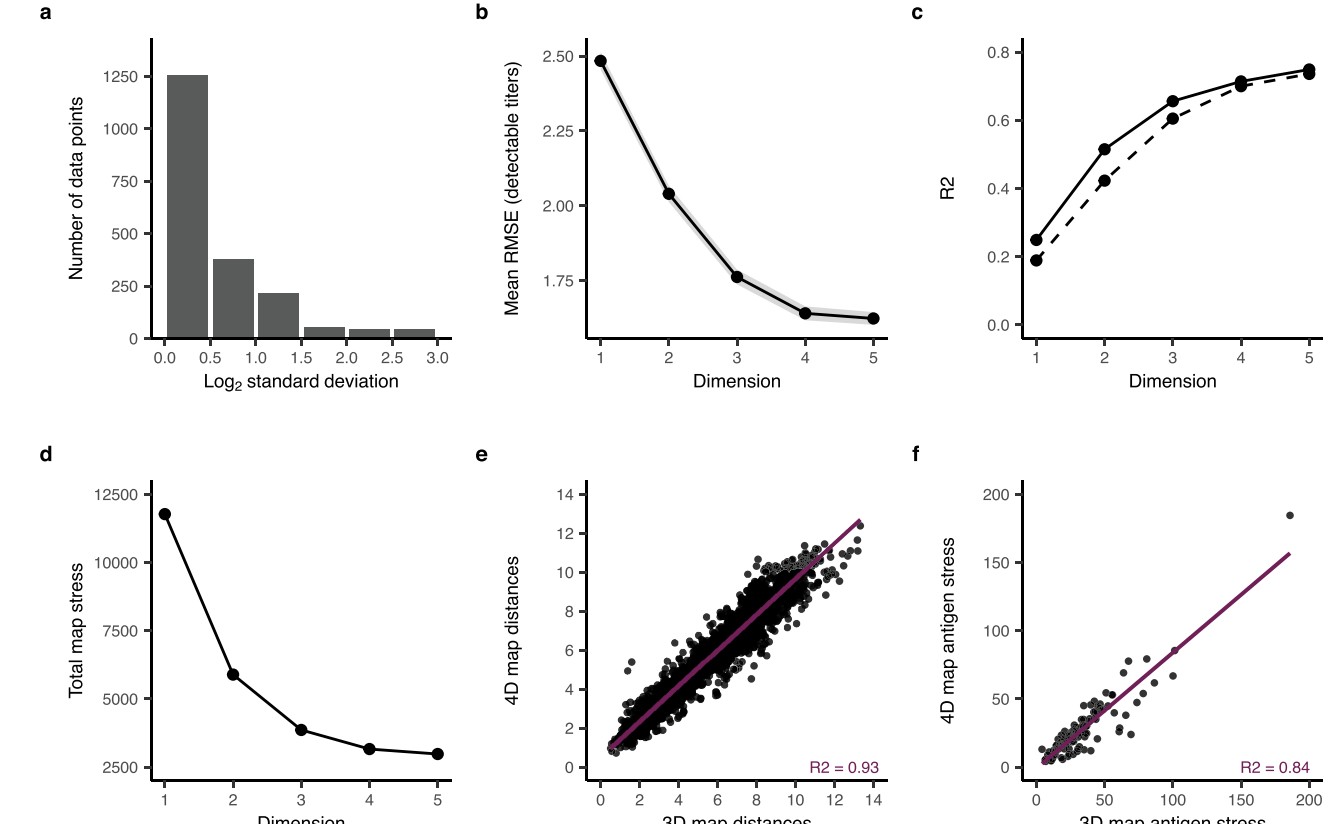

**Extended Data Fig. 1 | Analyses of A(H5) HI titre dataset variability and map dimensionality. a**, Histogram of standard deviation of $\log_2$ transformed HI titres between assays used to construct the full map dataset (127×33). Data points refer to unique antigen-serum combinations. **b**, Dimensionality test, indicating the mean root mean square error (RMSE) between predicted and measured detectable HI titres when 10% of the titres was removed, using 1 to 5 dimensions. The grey shading indicates the 95% confidence interval of the RMSE. **c**, Regression coefficient ($R^2$) of linear regression between HI titres (table distances), and corresponding Euclidian distances in the antigenic maps generated in one to five dimensions. The $R^2$ of the linear regressions based on all HI titres are shown as a solid line, and those including only detectable HI titres are shown as a dashed line. **d**, Total map stress using 1 to 5 dimensions. **e**, Scatter plots of pairwise antigen-serum map distances in the three- versus the four-dimensional map. **f**, Scatter plots of individual antigen stress in the three- versus the four-dimensional map. In e, f, linear regression lines are plotted, and the regression coefficient ($R^2$) is indicated. Analyses in b-f are performed using the 114×28 dataset (Supplementary Table 2).

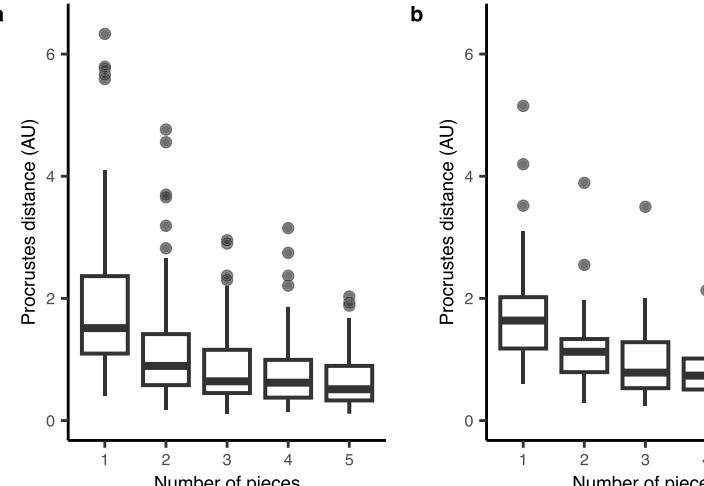

**Extended Data Fig. 2 | Piecewise Procrustes analysis comparing the A(H5) antigenic maps in three and four dimensions.** Procrustes distances between the three- and four-dimensional maps using one to five pieces for the piecewise Procrustes analysis are displayed. Box plots summarize the Procrustes distance for individual antigens (**a**) (n = 114), and sera (**b**) (n = 28). The line indicates the median, the lower and upper hinges the first and third quartiles, respectively, and the upper and lower whiskers extend respectively to the largest and smallest values that fall within 1.5 times the inter-quartile range (distance between the first and third quartiles). Data points beyond the end of the whiskers are plotted individually. Analyses were performed using the 114×28 dataset (Supplementary Table 2).

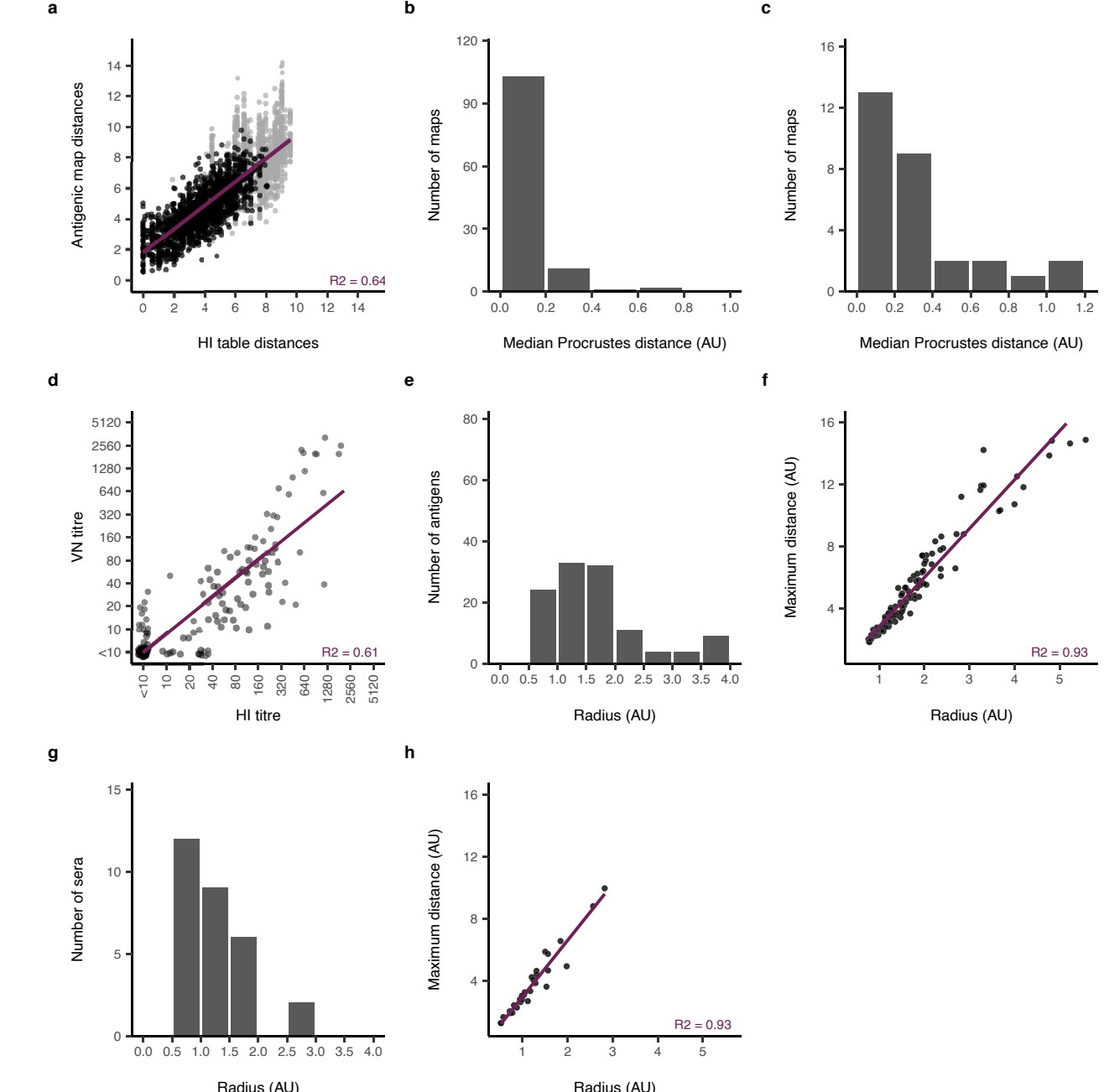

**Extended Data Fig. 3 | Accuracy and stability of the three-dimensional A(H5) antigenic map.** Analyses were performed using the final 117×29 dataset (Supplementary Table 2). **a**, Scatter plot of HI titre table distances versus the corresponding Euclidian distances in the antigenic map distances inferred from detectable titres (black) and from non-detectable titres (grey) are shown. **b-c**, Histograms showing the distribution of the median Procrustes distance over individual maps as compared with the full antigenic map upon removing individual antigens (b) or sera (c) from the antigenic map. **d**, Scatter plot of HI titres versus virus neutralization (VN) titres. **e-g**, Histograms of the radius of Bayesian bootstrap blobs for antigens (e) and sera (g). **f-h**, Comparison the maximum width of Bayesian bootstrap blobs versus the radius of equal-volume spheres for antigens (f) and sera (h). In a, d, f, h, linear regression lines are plotted, and the regression coefficient ($R^2$) is indicated.

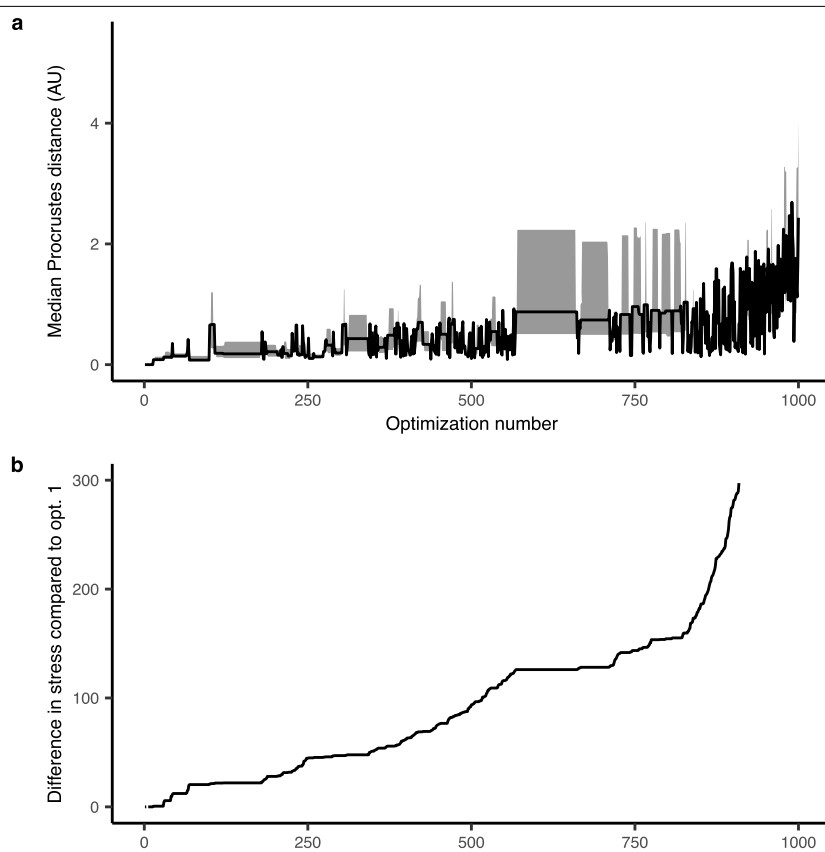

**Extended Data Fig. 4 | Analyses of the A(H5) antigenic map stability across 1000 ranked optimizations.** Analyses were performed using the final 117 × 29 dataset (Supplementary Table 2). **a**, Median Procrustes distance between the position of points in each optimization as compared with those in the lowest stress optimization (opt. 1) for ranked optimizations in increasing stress orders. Shown is the median Procrustes distance (black line), and the distribution of Procrustes distances excluding the highest and lowest 20% (grey shading). **b**, Difference in total map stress as compared with the lowest stress optimization (opt. 1) for all 1000 optimizations. Values above 300 (beyond optimization 910) were not plotted to allow a detailed visualization of the values below 300. The maximum value observed was 1129 for optimization 1000. Opt.: optimization.

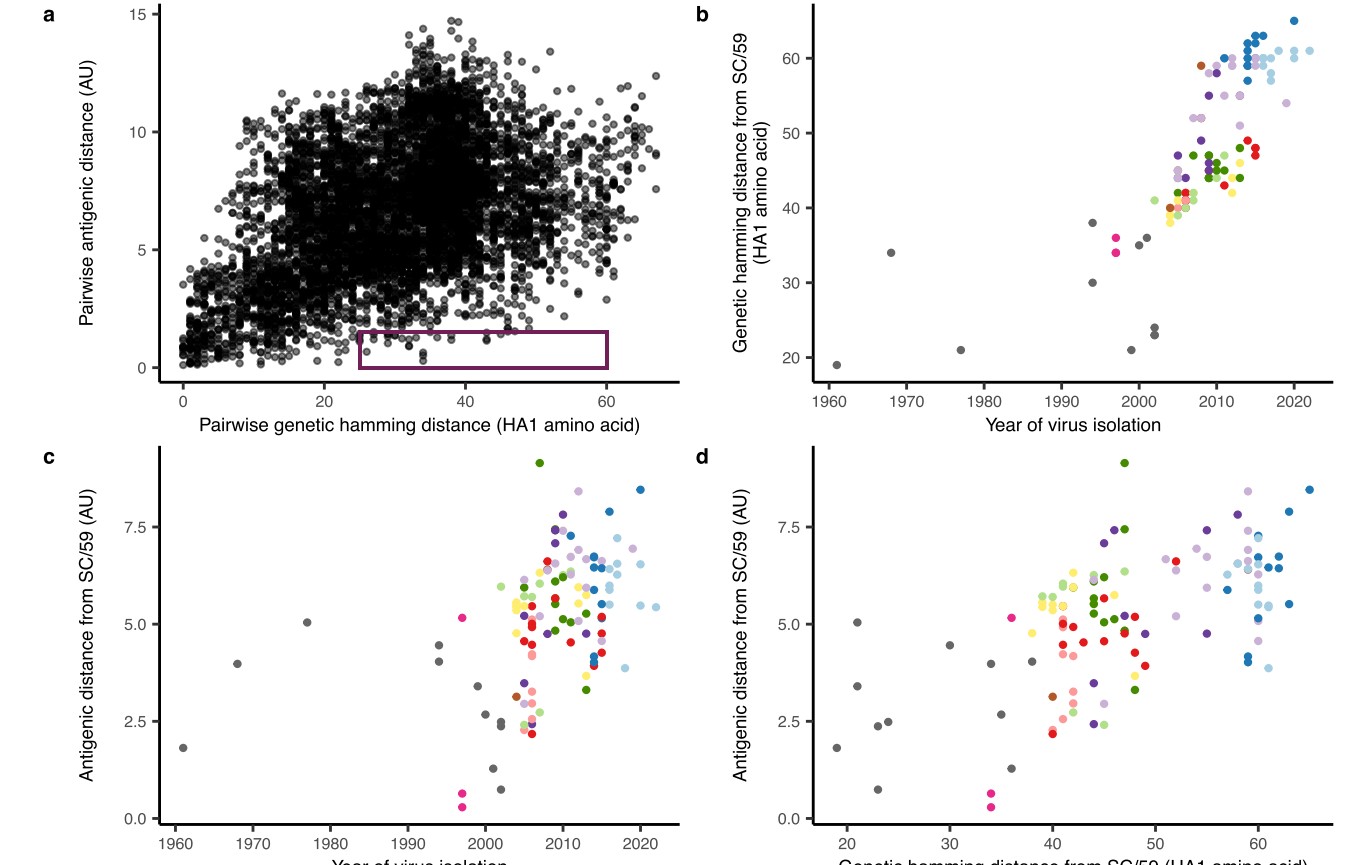

**Extended Data Fig. 5 | Genetic and antigenic diversity of antigens in the A(H5) antigenic map. a**, Pairwise antigenic distances derived from the antigenic map (117 × 29) as a function of pairwise genetic hamming distances (amino acid differences between two antigens) shown for all antigen-antigen pairs. The rectangle highlights the selected outlier antigen-antigen pairs with a genetic distance above 25 amino acids and an antigenic distance below 1.5 AU. Points are depicted in grey including a degree of transparency allowing the visualization of overlapping points. **b**, Genetic hamming distance from A/chicken/Scotland/1959, first A(H5) virus ever isolated, as a function of the year of virus isolation, shown for each antigen in the map. **c**, Antigenic distance from A/chicken/Scotland/1959 as a function of the year of virus isolation, shown for each antigen in the map. **d**, Antigenic distance from A/chicken/Scotland/1959 as a function of genetic hamming distance from A/chicken/Scotland/1959, shown for each antigen in the map. In b-d, the colour of each point corresponds to the antigen's genetic clade as indicated in Fig. 1b. SC/59: A/chicken/Scotland/1959.

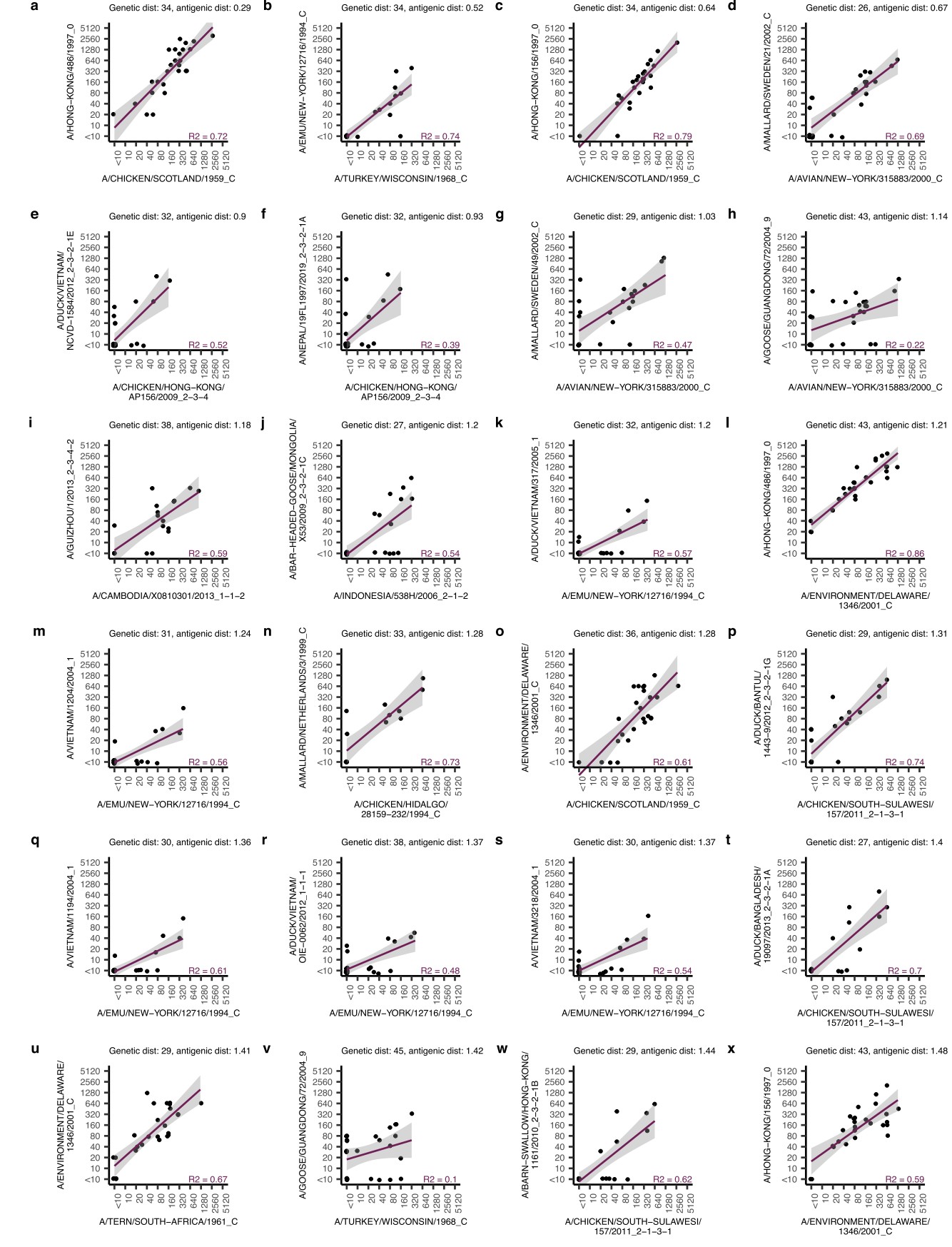

**Extended Data Fig. 6 |** See next page for caption.

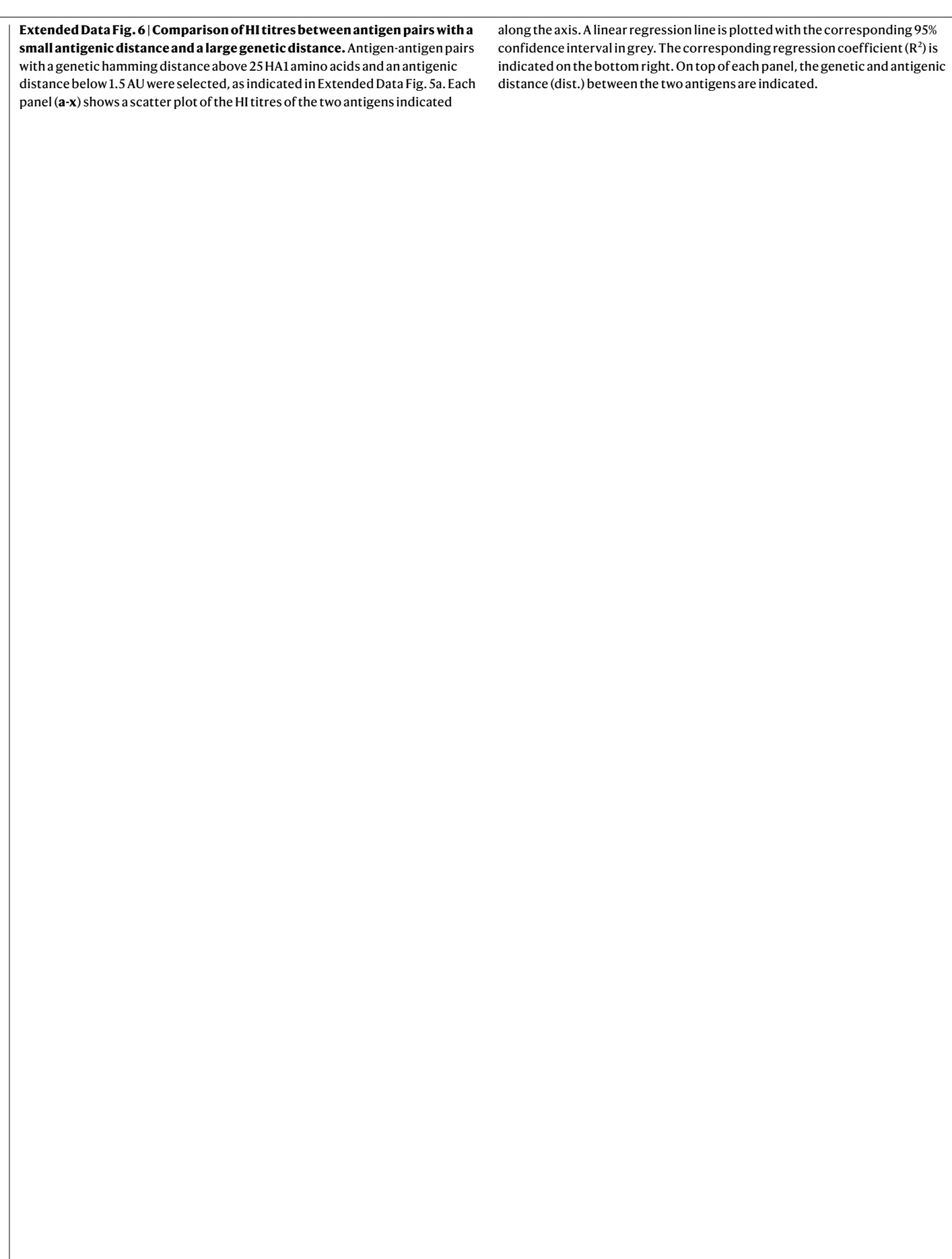

**Extended Data Fig. 6 | Comparison of HI titres between antigen pairs with a small antigenic distance and a large genetic distance.** Antigen-antigen pairs with a genetic hamming distance above 25 HA1 amino acids and an antigenic distance below 1.5 AU were selected, as indicated in Extended Data Fig. 5a. Each panel (**a-x**) shows a scatter plot of the HI titres of the two antigens indicated along the axis. A linear regression line is plotted with the corresponding 95% confidence interval in grey. The corresponding regression coefficient ($R^2$) is indicated on the bottom right. On top of each panel, the genetic and antigenic distance (dist.) between the two antigens are indicated.

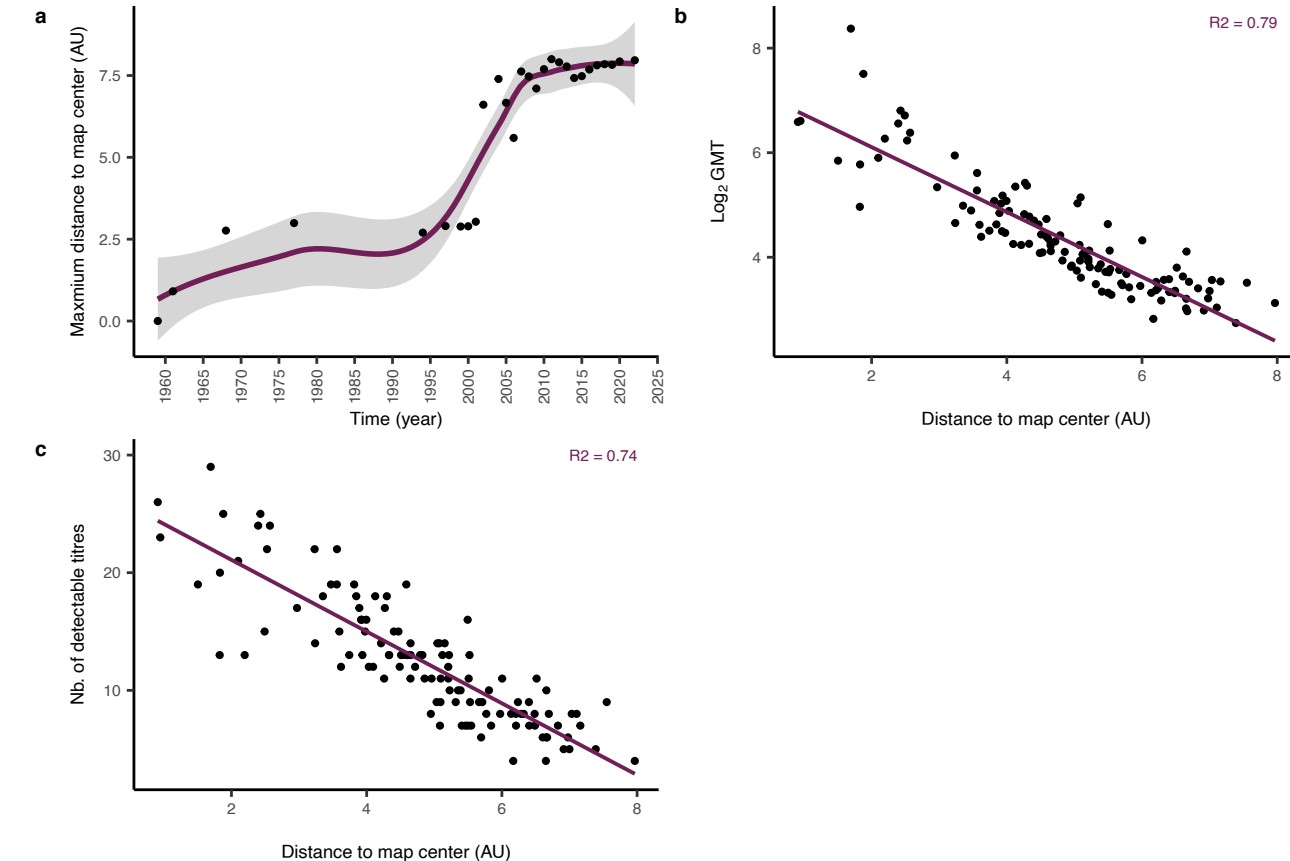

**Extended Data Fig. 7 | Antigen distance to the centre of the map.** The map centre corresponds to the mean of all antigen position coordinates in each direction (x, y and z). Analyses were performed using the final 117×29 dataset (Supplementary Table 2). **a**, Maximum antigen distance from the map centre over time. For each year in the dataset, the positions of all antigens from viruses isolated up until that year were used to compute the centre of mass. The maximum distance of the respective antigens to the centre was plotted. A smoothed local regression curve is plotted with the 'geom_smooth' function (ggplot2) using method = 'loess'. The 95% confidence interval is displayed in grey. **b**, $Log_2$ transformed geometric mean titre as a function of the distance to the map centre in antigenic units shown for each antigen in the map. **c**, Number of detectable HI titres in the dataset as a function of the distance to the map centre in antigenic units shown for all antigens in the map. In b and c, linear regression lines are plotted, and the regression coefficient ($R^2$) is indicated.

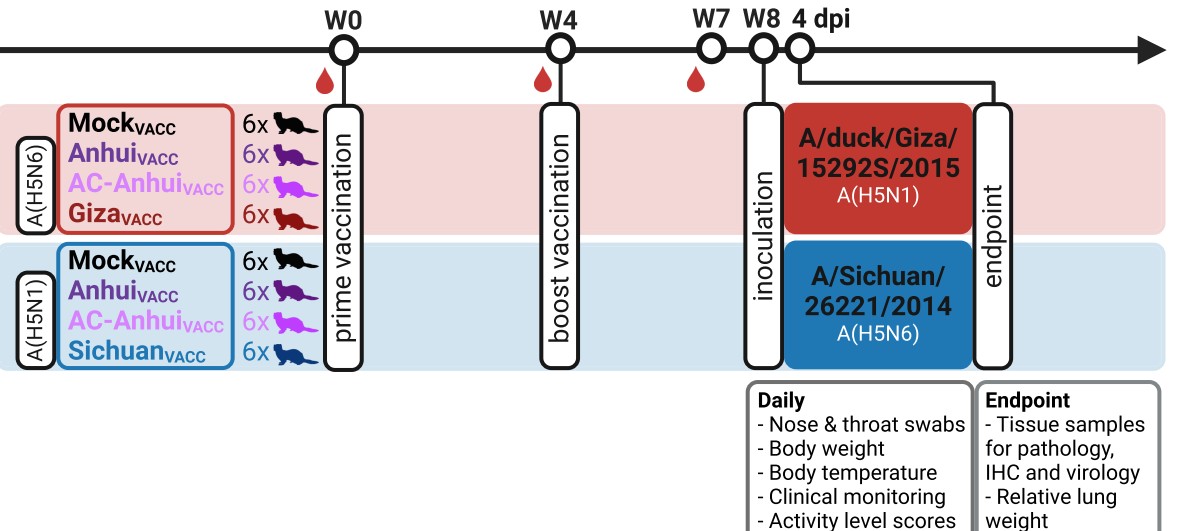

**Extended Data Fig. 8 | Vaccination-challenge study design.** Schematic overview of the two vaccination-challenge experiments. The timeline in weeks (W) is indicated at the top. Six ferrets per group were vaccinated twice with adjuvanted split-inactivated vaccines four weeks apart. Red drops denote serum sample collection before the first and second vaccination, as well as three weeks after the second vaccination. Four weeks after the boost vaccination, animals were inoculated either with A/duck/Giza/15292S/2015 virus (A(H5N1), clade 2.2.1.2) or with A/Sichuan/26221/2014 virus (A(H5N6), clade 2.3.4.4a). Daily, nose and throat swabs were collected, and body weight, body temperature, clinical signs, and activity level scores were monitored. Ferrets were sacrificed four days post-inoculation, and relevant tissues were collected for virological and histopathological analysis. W: week. Created in BioRender. 3, V. (2025) https://BioRender.com/efgr2hr.

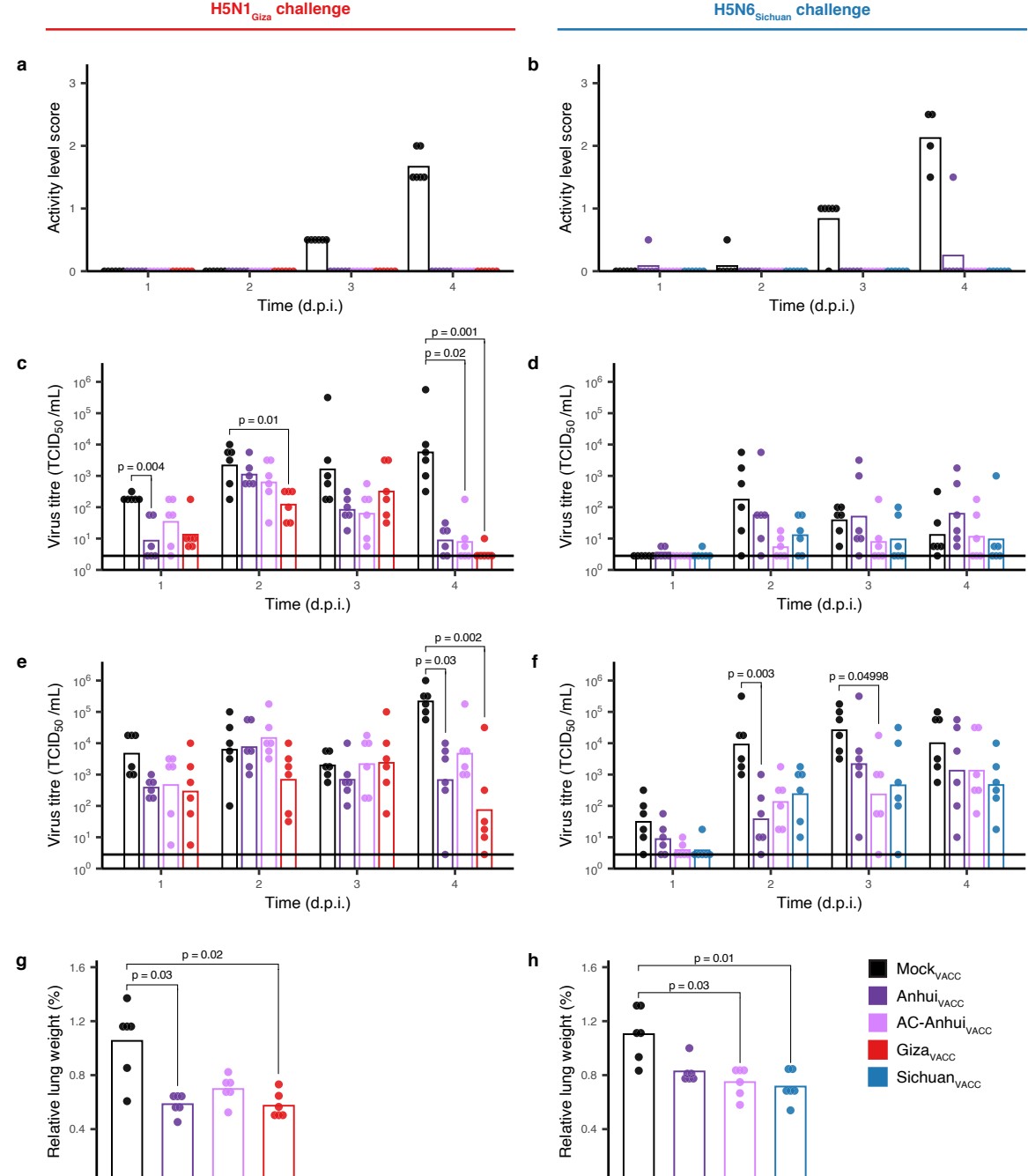

**Extended Data Fig. 9 | Activity level scores, virus titres in respiratory swabs and relative lung weight of ferrets in vaccination-challenge studies.** Vaccination-challenge with the H5N1$_{Giza}$ virus (left panels), and the H5N6$_{Sichuan}$ virus (right panels). **a-b,** Daily activity level scores determined as follows: 0 - alert and playful, 1 - alert and playful only when stimulated, 2 - alert but not playful when stimulated, 3 - neither alert nor playful when stimulated. Bars represent the group mean (n = 6), and dots represent the activity level score of individual animals. **c-f,** Virus titres (TCID$_{50}$/mL) in nose (c and d) and throat swabs (e and f) on a logarithmic scale. Bars represent the group geometric mean titre (n = 6) and dots represent the titres in swabs of individual animals. The horizontal lines indicate the detection limit. Virus titrations were performed in four replicates. **g-h,** Relative lung weight, defined as the percentage of lung weight relative to total body weight at necropsy. Bars represent the mean per group (n = 6), and dots represent the relative lung weight of individual animals. Significant differences between groups were assessed with a Kruskal-Wallis test followed by a pairwise two-sided Dunn's test with Bonferroni correction for multiple comparisons. All pairwise comparisons were tested. Only statistically significant differences (p < 0.05) are indicated with the corresponding p-value in panels (**c-h**). All graphs are colour-coded according to the experimental groups, as indicated in the legend.

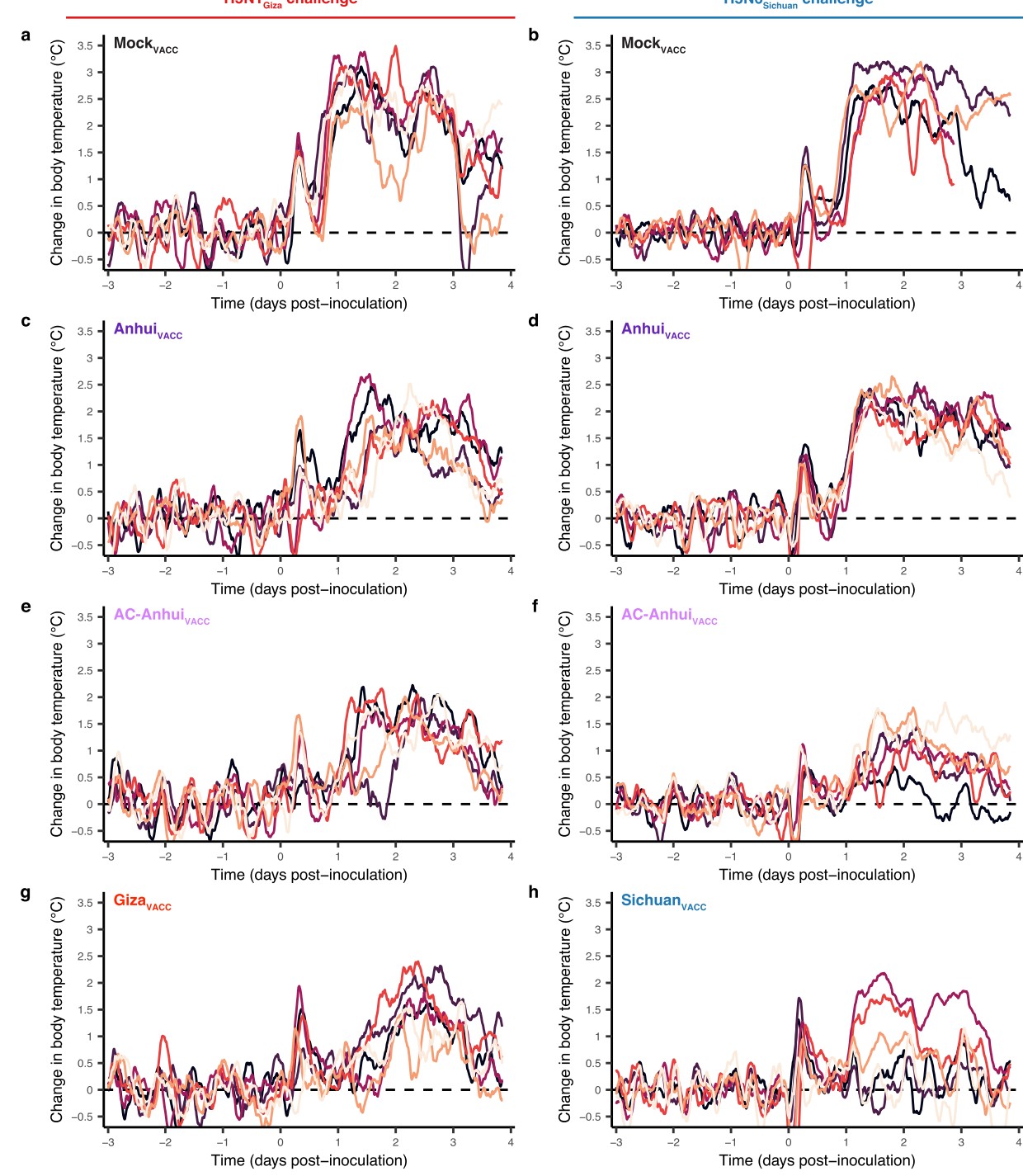

**Extended Data Fig. 10 | Body temperature change of individual ferrets in the vaccination-challenge studies.** Vaccination-challenge with the H5N1$_{Giza}$ virus (left panels): **a**, Mock$_{VACC}$; **c**, Anhui$_{VACC}$; **e**, AC-Anhui$_{VACC}$; and **g**, Giza$_{VACC}$ groups. Vaccination-challenge with the H5N6$_{Sichuan}$ virus (right panels): **b**, Mock$_{VACC}$, **d**, Anhui$_{VACC}$, **f**, AC-Anhui$_{VACC}$, and **h**, Sichuan$_{VACC}$. Body temperature change from baseline (mean body temperature recorded during the three days prior to inoculation, indicated as horizontal black line). Body temperature was recorded every ten minutes using probes surgically implanted in the peritoneal cavity. Four-hour sliding means are displayed. Each line represents the data of an individual animal (n = 6). In the H5N6$_{Sichuan}$ challenge Mock$_{VACC}$ group (b), data of five individual animals are shown due to a malfunctioning temperature probe, and data of the two deceased animals are included up until three days post-inoculation.

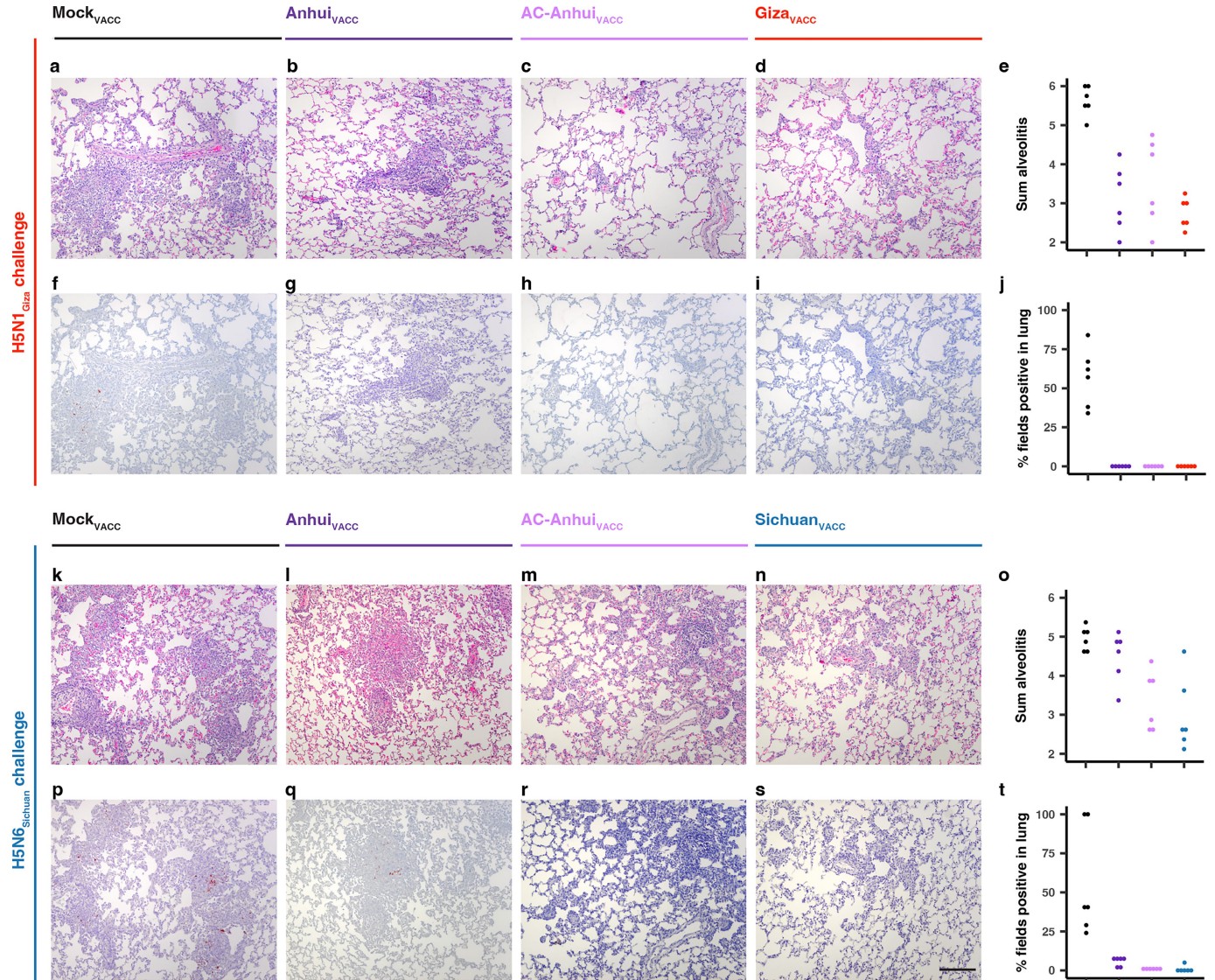

**Extended Data Fig. 11 | Histopathology and immunohistochemistry of lung tissues from ferrets in the vaccination-challenge studies.** Representative images of formalin-fixed paraffin-embedded lung sections from animals with representative scores for histopathology and influenza virus nucleoprotein (NP) antigen expression for each experimental group are shown. (**a-d** and **k-n**) Lung sections stained with hematoxylin and eosin. **e** and **o**, Sum of the scoring of severity and extent of alveolitis for each animal (n = 6). Severity of alveolitis was scored as follows: 0: no inflammatory cells; 1: few inflammatory cells; 2: moderate number of inflammatory cells; 3: many inflammatory cells.

Extend of alveolitis was scored as follows: 0: 0%; 1: 1–25%; 2: 25–50%; 3: >50%. (**f-i** and **p-s**) Immunohistochemical staining of lung sections with anti-influenza NP antibody. (**j** and **t**) Scoring of percentage of NP-positive fields for each animal (n = 6). Cumulative scores for each animal are the percentage of fields positive for NP based on 25 arbitrarily chosen, 20x objective, fields of lung parenchyma. Vaccination-challenge studies and vaccine groups are indicated in the figure. Data are colour-coded based on vaccine group as indicated in the figure. A 200 μm scale bar for panels a-d, k-n, f-i and p-s is shown in s.

# Reporting Summary

## Statistics

For all statistical analyses, confirm that the following items are present in the figure legend, table legend, main text, or Methods section.

| n/a | Confirmed | |
|---|---|---|
| ☐ | ☒ | The exact sample size (*n*) for each experimental group/condition, given as a discrete number and unit of measurement |
| ☐ | ☒ | A statement on whether measurements were taken from distinct samples or whether the same sample was measured repeatedly |
| ☐ | ☒ | The statistical test(s) used AND whether they are one- or two-sided *Only common tests should be described solely by name; describe more complex techniques in the Methods section.* |
| ☒ | ☐ | A description of all covariates tested |
| ☐ | ☒ | A description of any assumptions or corrections, such as tests of normality and adjustment for multiple comparisons |
| ☐ | ☒ | A full description of the statistical parameters including central tendency (e.g. means) or other basic estimates (e.g. regression coefficient) AND variation (e.g. standard deviation) or associated estimates of uncertainty (e.g. confidence intervals) |
| ☐ | ☒ | For null hypothesis testing, the test statistic (e.g. *F*, *t*, *r*) with confidence intervals, effect sizes, degrees of freedom and *P* value noted *Give P values as exact values whenever suitable.* |
| ☒ | ☐ | For Bayesian analysis, information on the choice of priors and Markov chain Monte Carlo settings |
| ☒ | ☐ | For hierarchical and complex designs, identification of the appropriate level for tests and full reporting of outcomes |
| ☒ | ☐ | Estimates of effect sizes (e.g. Cohen's *d*, Pearson's *r*), indicating how they were calculated |

*Our web collection on statistics for biologists contains articles on many of the points above.*

## Software and code

Policy information about availability of computer code

| Data collection | Cell Olympus for micrographs acquisition. |
|---|---|
| Data analysis | R version 4.4.3. Python version 3.10.14. R package Racmacs, available at: https://acorg.github.io/Racmcas/ (version 1.2.3). Custom R and Python scripts were used, which are publicly available at https://github.com/epiv-lab/H5-antigenic-evolution (zenodo: https://doi.org/10.5281/zenodo.13237524) and https://github.com/epiv-lab/pepiniere (zenodo: https://doi.org/10.5281/zenodo.12751132). Python trimesh package (version 3.2.0). Python PyRacmacs package, available at https://github.com/iAvicenna/PyRacmacs. MAFFT version v7.515. IQ-Tree2 version 2.1.4_beta. ggtree version 1.4.11. LABEL (H5v2023 pre-release 1 (2023-05-05)). R Biostring package version 2.74.1. R rj3s package version 0.0.2. R ggplot2 package version 3.5.1. R flexdashboard package version 0.6.2. Webshot version 0.1.2. |

For manuscripts utilizing custom algorithms or software that are central to the research but not yet described in published literature, software must be made available to editors and reviewers. We strongly encourage code deposition in a community repository (e.g. GitHub). See the Nature Portfolio guidelines for submitting code & software for further information.

## Data

Policy information about availability of data

All manuscripts must include a data availability statement. This statement should provide the following information, where applicable:
- Accession codes, unique identifiers, or web links for publicly available datasets
- A description of any restrictions on data availability
- For clinical datasets or third party data, please ensure that the statement adheres to our policy

All data are available in the main text or the supplementary materials. Sequence accession numbers are available in Supplementary Table 1. Supplementary Data 1 to 10 are available via https://epiv-lab.github.io/H5-antigenically-central-vaccine/.

## Research involving human participants, their data, or biological material

Policy information about studies with human participants or human data. See also policy information about sex, gender (identity/presentation), and sexual orientation and race, ethnicity and racism.

| Reporting on sex and gender | Research does not involved human participants or human data. |
|---|---|
| Reporting on race, ethnicity, or other socially relevant groupings | Research does not involved human participants or human data. |
| Population characteristics | Research does not involved human participants or human data. |
| Recruitment | Research does not involved human participants or human data. |
| Ethics oversight | Research does not involved human participants or human data. |

Note that full information on the approval of the study protocol must also be provided in the manuscript.

# Field-specific reporting

Please select the one below that is the best fit for your research. If you are not sure, read the appropriate sections before making your selection.

☒ Life sciences          ☐ Behavioural & social sciences          ☐ Ecological, evolutionary & environmental sciences

For a reference copy of the document with all sections, see nature.com/documents/nr-reporting-summary-flat.pdf

# Life sciences study design

All studies must disclose on these points even when the disclosure is negative.

| Sample size | The use of 6 animals in vaccination challenge experiments makes it possible to demonstrate relevant differences between test groups (de Wit et al., J Virol, 2005, 79 (19): 12401) (Richardson, J Virol., 2005, 79 ( 2): 669-676) (Kreijtz et al, Vaccine, 2007). The current experimental set-up allows us to compare differences in viral replication between the vaccination groups. In addition, historical data concerning the difference and the spread between groups allows us to also estimate the group size:<br>$n = 2'[\{(z_{1-a} / 2 + z_{1-b}) s\} / (m_1 - m_0)]^2$<br>$a = 0.05$<br>$1-b = 0.8$ (Power of 80)<br>$s = 2.46$ (spread between groups in log units)<br>$m_1 = 3.52$ (difference between groups in log units)<br>$m_0 = 0$<br><br>--> n=6 |
|---|---|
| Data exclusions | No data were excluded from analyses |
| Replication | In vivo experiments were replicated 6 times (see above justification of a group size of 6 for vaccination-challenge experiments. A subset of the hemagglutination inhibition titrations were replicated twice or more to evaluate assay variation (presented in Ext. Figure 1A). Titrations of the ferret swabs and tissues were performed in quadriplicates. |
| Randomization | Animals were randomly allocated to the different groups. Randomization was not applicable to the other experiments. |
| Blinding | Animal experiments were not blinded to the investigators due to regulations that require knowledge of animal treatment for biosafety reasons. Pathological scoring and analyses were performed blindly. Titrations and hemagglutination inhibition assays were read out blindly. |

# Reporting for specific materials, systems and methods

We require information from authors about some types of materials, experimental systems and methods used in many studies. Here, indicate whether each material, system or method listed is relevant to your study. If you are not sure if a list item applies to your research, read the appropriate section before selecting a response.

## Materials & experimental systems

| n/a | Involved in the study |
|---|---|
| ☐ | ☒ Antibodies |
| ☐ | ☒ Eukaryotic cell lines |
| ☒ | ☐ Palaeontology and archaeology |
| ☐ | ☒ Animals and other organisms |
| ☒ | ☐ Clinical data |
| ☒ | ☐ Dual use research of concern |
| ☒ | ☐ Plants |

## Methods

| n/a | Involved in the study |
|---|---|
| ☒ | ☐ ChIP-seq |
| ☒ | ☐ Flow cytometry |
| ☒ | ☐ MRI-based neuroimaging |

## Antibodies

| | |
|---|---|
| Antibodies used | mouse IgG2a anti-influenza A nucleoprotein, H16-L10-4R5 (ATCC® HB-65™, RRID: CVCL_4524); mouse IgG2a isotype control (R&D, MAB003, RRID:AB_357345); goat anti-mouse IgG2a secondary antibody coupled to horseradish peroxidase (HRP) (Biorad, Star133A, RRID: AB_1102655P). |
| Validation | The mouse IgG2a anti-influenza A nucleoprotein was tested by testing several dilutions on a tissue with known positivity (ferret nasal turbinates and cat lung infected with pH1N1. The lowest dilution of antibody leading to a clear signal (red precipitate) was chosen. Specificity was assessed using an isotype control, for which the same amount of antibody as the primary antibody was used. |

## Eukaryotic cell lines

Policy information about cell lines and Sex and Gender in Research

| | |
|---|---|
| Cell line source(s) | HEK293T and MDCK cells were acquired from ATCC |
| Authentication | None of the cell lines used were authenticated |
| Mycoplasma contamination | The cell banks were tested negative for mycoplasma. |
| Commonly misidentified lines (See ICLAC register) | None |

## Animals and other research organisms

Policy information about studies involving animals; ARRIVE guidelines recommended for reporting animal research, and Sex and Gender in Research

| | |
|---|---|
| Laboratory animals | Mustela putorius furo, 6 to 1 year old |
| Wild animals | No wild animals were used in this study |
| Reporting on sex | males (sera production) and females (vaccination challenge experiment) |
| Field-collected samples | No field-collected samples were used |
| Ethics oversight | Ferret experiments were performed in strict compliance with the Dutch legislation on the protection of animals used for scientific purposes (2014, European Union directive 2010/63/EU implemented). Experiments were performed at the Erasmus Medical Center in Rotterdam, the Netherlands under a project license accredited by the Dutch competent authority (license number AVD101002015340). Study protocols were approved by the Erasmus Medical Center Animal Welfare Body (permit numbers 15-340-01, -04, -06, -22, -23, and -24). |

Note that full information on the approval of the study protocol must also be provided in the manuscript.

# Plants

Seed stocks

*Report on the source of all seed stocks or other plant material used. If applicable, state the seed stock centre and catalogue number. If plant specimens were collected from the field, describe the collection location, date and sampling procedures.*

Novel plant genotypes

*Describe the methods by which all novel plant genotypes were produced. This includes those generated by transgenic approaches, gene editing, chemical/radiation-based mutagenesis and hybridization. For transgenic lines, describe the transformation method, the number of independent lines analyzed and the generation upon which experiments were performed. For gene-edited lines, describe the editor used, the endogenous sequence targeted for editing, the targeting guide RNA sequence (if applicable) and how the editor was applied.*

Authentication

*Describe any authentication procedures for each seed stock used or novel genotype generated. Describe any experiments used to assess the effect of a mutation and, where applicable, how potential secondary effects (e.g. second site T-DNA insertions, mosiacism, off-target gene editing) were examined.*

