## [Peer Review file · Nature]

A vaccine central in A(H5) influenza antigenic space confers broad immunity

Corresponding Author: Dr Mathilde Richard

Version 0:

Reviewer comments:

Referee #1

(Remarks to the Author)

In their manuscript Kok and colleagues describe a novel vaccine approach for HPAI H5N1 that is based on an antigenically centralized HA. The approach is interesting and the work is well done. However, there are several points that need the authors' attention.

Major points

- 1) The approach is different than previous centralized approaches which were often sequence-based and not necessarily based on antigenic data. However, the authors are pretty dismissive of these previous approaches which have actually worked well too.
- 2) It would have been good to use another unrelated inactivated influenza virus as negative control, e.g. and influenza B virus preparation with adjuvant.

Minor points

- 1) Line 39: HA1 also contains part of the stalk.
- 2) Line 45: In Australia too?
- 3) Figure 1B: It would be better to label the clades within the figure, not on the side.
- 4) Line 112: Are really the non-gooseGd lineages meant?
- 5) Line 115: 'A zoomable pdf file...'
- 6) Line 161 and 162: Remove 'it' twice.
- 7) Line 181 and throughout the manuscript: It is 'alpha2.6-linked sialic acid' not 'alpha 2.6 sialic acid'. Same for 2.3 of course.
- 8) Line 206: Define 'GMT'.
- 9) Line 216: Please correct the strain name.
- 10) Line 283: Please indicate the challenge dose for the respective viruses.
- 11) Line 283 and throughout the manuscript: Please use proper strain names or abbreviations but not 'Giza' and 'Sichuan'.

- 12) Line 300: 'trachea', not 'tracheas'
- 13) Line 422, line 631: 'et al.' should be in italics.
- 14) Line 440: The only published human studies with mRNA vaccines for avian influenza viruses (H7 and H10) do actually indicate worse immunogenicity of this platform compared to inactivated adjuvanted vaccines.
- 15) Many abbreviations in the methods section are not defined.
- 16) Line 510: It should be 'Dulbecco's'.
- 17) Line 513 and 516: Streptomycin is typically not given in U/ml but ug/ml.
- 18) Line 513: 'geneticin', not 'Geneticin'
- 19) Line 617: Please describe for the reader what 'Mega10' is.
- 20) Line 733: Please use non-capitalized letters to start words mid-sentence.
- 21) Line 769: Where is the data from the VN assays used?

Referee #2

(Remarks to the Author)

Kok et al. selected 33 viruses that represented the genetic diversity of H5 influenza viruses. Using the gold-standard ferret model of influenza pathogenesis and vaccination, anti-sera was raised against these viruses and tested for cross-reactivity via hemagglutination inhibition (HI) assay to 127 different H5 antigens. After excluding 4 anti-sera and 10 antigens due to poor reactivity, the results of the cross-HI assays were used to perform antigenic cartography. This analysis revealed discordance between genetic and antigenic evolution. The antigenic map further showed that antigenic evolution did follow a specific pattern or was non-directional. From the antigenic map, several central H5 antigens were chosen for further evaluation as candidate vaccines. Mutations were introduced into the HA to alter receptor-binding preference and remove a putative glycosylation site. Ferret anti-sera was then raised against candidate vaccine antigens, and after performing cross-HI assays, the A/Anhui/1/2005 (H5N1) HA was chosen as the most suitable antigen. The stability of the A/Anhui/1/2005 HA was then evaluated during repeated passaging in eggs and sequencing of the passaged viruses revealed that the 134TA mutation was selected. This mutation was then incorporated into the final vaccine antigen. Last, to evaluate vaccine efficacy, vaccination and challenge studies were performed using the candidate vaccine and two heterologous challenge strains. Vaccines matching the challenge strains were also incorporated as comparators. After vaccination and challenge, ferrets vaccinated with the A/Anhui candidate vaccine were protected from disease and the level of protection was comparable to control vaccines matching the challenge strain. Collectively, these studies demonstrate that antigenic cartography can be used to select vaccine antigens with enhanced breadth of protection.

Comments:

The use of antigenic cartography to select a central H5 antigen represents an advancement in the design of H5 vaccines. Moreover, the experimental studies represent a large body of research, and the findings demonstrate the central H5 antigens with additional mutations induce a cross-reactive HI response. The findings are presented in detail with multiple supplemental tables and figures. The statistical analyses are appropriate and the conclusions and stated limitations are accurate.

There are sections of the manuscript that would benefit from revisions for clarity. Specifically, lines 77-93, indicate that 127 antigens were selected, and 33 anti-sera were generated based on preliminary studies. However, it would be much easier to follow if the process was instead explained as 33 anti-sera were generated and tested for cross-reactive to 127 HAs. Line 97 indicates "four detectable titers", is this meant to convey 4 HI units. Please clarify. Parts of the manuscript indicate the H5 antigens were engineered or rationally designed (see lines 28, 410, etc). This is slightly misleading as central antigens were chosen and mutations were introduced. The rationale for introducing the mutations is justified; however, the terms "rationally designed or engineered antigens", at least to this reviewer, imply a completely synthetic approach rather than selecting from existing viruses. Extended Data Figure 1A, please clarify what is meant by "number of titers" on the y-axis.

In the virus challenge studies, the authors indicate the dose used was determined in preliminary studies. Please add details to clarify if the challenge dose was lethal. Also in the methods section, please indicate the method of euthanasia.

While the manuscript uses antigen cartography which is an advanced approach, studies such as those by Boonnak et al., (J Virology 2017) in which anti-serum to several H5 viruses was tested for cross-reactivity to select a broadly cross-reactive H5 vaccine strain should be referenced and discussed. It would be particularly useful to add details of how antigen cartography is an improvement over cross-HI alone.

The manuscript refers extensively to supplemental tables and figures. This is due to the complex nature of the dataset. Within the supplemental tables, it would be very helpful if vaccine and challenge strains were highlighted in bold. Within the antigenic maps, it would similarly be helpful, if the vaccine and challenge strains were denoted with symbols that made them readily identifiable from all other viruses. This would allow the reader to quickly identify these strains and assess their relationship to other viruses.

Referee #3

(Remarks to the Author)

Kok and colleagues present a very nice series of studies designing and testing new H5 vaccine antigens. They used ferret antisera to generate an H5 antigenic map and then identified viral isolates that were centrally located in the antigenic map. They then modified centrally located H5 antigens by removing a glycosylation site and adding mutations that facilitate α 2-6 sialic acid binding. A split vaccine was created using one of these modified H5s—this vaccine (along with the unmodified central antigen) protected ferrets against challenge with 2 distinct H5 strains—and protection was similar to that afforded by homologous vaccines. The paper is important and presents an interesting concept, but it is not clear if the modified vaccine provides better protection compared to other mismatched antigens. Regardless, I think that this is a tremendously valuable set of experiments and it will move the field forward.

1. This is an interesting idea and the manuscript is filled with interesting and useful data. But it is not clear that the modified vaccine (AC-Anhui-VACC) is much better than the unmodified centrally located antigen (Anhui-VACC). It is not clear that this vaccine is much better than a different vaccine antigen that is not centrally located on the antigenic maps, because this type of non-centrally located antigen was not included in the vaccine challenge studies. It is possible that most of the protection in the challenge studies is due to non-neutralizing antibodies and that most H5 antigens would afford some level of protection in these experiments. All of the vaccines tested protected against Giza and there were only slightly small differences between AC-Anhui and Anhui with protection against Sichuan.
2. The authors modified the centrally located HA antigens by introducing mutations that directly altered receptor binding specificity. The authors completed experiments to document how these mutations affected specificity. It is possible that these mutations are also changing the overall avidity and HA binding strength to the turkey red blood cells used in HI assays. It is possible that these mutations decrease overall HA binding to turkey red blood cells. Viruses that bind poorer to red blood cells can have inflated HI titers, simply due to differences in binding avidity rather than differences in antigenicity. The authors should consider measuring overall avidity to determine if the higher HI values with the modified HAs is due to weaker red blood cell binding.
3. Ferrets tend to produce antibodies mainly against the top of HA that target residues near the RBS. Human typically produce antibodies that target multiple different sites, and it is possible that antigenic maps based on ferret serum are underestimating the contribution of antigen sites further from the RBS. Presumably, the loss of a glycan at HA residue 154 increases the accessibility of the top of HA, which is the immunodominant site in ferrets. Are the RBD and glycosylation patterns of Giza and Sichuan different, and do these viruses share homology with the modified vaccine antigen in the RBD? Does the modified vaccine elicit antibodies against the immunodominant top of HA that is similar in Giza and Sichuan?
4. The HI tables in Supplemental tables 8 and 9 are important and should be incorporated in some manner into the main figures. I understand these data are the basis of the antigenic maps, but is there an abbreviated table or figures that can make it easy for the reader to directly compare titers (for example, direct comparison of antibodies elicited by Anhui-VACC versus AC-Anhui-VACC against individual viral strains to evaluate differences)?
5. Does 134TA lead to a glycan loss?

Version 1:

Reviewer comments:

Referee #1

(Remarks to the Author)

The authors addressed the reviewers' comments well.

Referee #2

(Remarks to the Author)

Kok et al. have submitted a revised version of their manuscript which seeks to use antigenic cartography to identify an antigenically central H5 HA antigen for use as a broadly H5 virus protective vaccine. The revised manuscript has addressed several previous comments; however, there are still some remaining comments to consider/address.

Comments:

- 1) The vaccines tested in the manuscript are adjuvanted split-virion vaccines, and the adjuvant combined with modifications to the HA are likely contributing to the breadth of the antibody response. It is unclear if the vaccines were not adjuvanted if a central H5 HA antigen would have similar protection as a homologous vaccine. In the event of a pandemic, while adjuvants

would certainly be beneficial, it is unknown if they would be approved for use in humans. For the 2009 H1N1 pandemic, adjuvanted vaccines were approved in Europe but not the United States. This should be addressed either by adding rationale in the introduction or results for including an adjuvant (i.e. indicating that H5 vaccines without an adjuvant are unlikely to be immunogenic), or by listing the use of an adjuvant as a limitation in the discussion and that additional studies without an adjuvant are warranted.

2) The description of the statistical approach in the methods and figures is inconsistent, and the statistical approach requires revisions. In the methods, the authors indicate a Kruskal-Wallis test was first performed to determine statistical significance, and this was followed by pairwise Mann-Whitney U tests. There are other post-hoc tests such as a Dunn's test that are more appropriate than pairwise Mann-Whitney U tests. Mann-Whitney U tests are meant for pair-wise comparisons and do not account for multiple comparisons. If Mann-Whitney U tests are used, then the full statistical approach needs to be clarified in the figure legends. It is also not accurate to denote with a single horizontal bar that 3 or 4 groups differ from Mock. Instead, each individual comparison should be depicted, or it should be clarified in the figure legend that only the comparison between the mock and each individual group were performed and the line does not imply the other groups are not different from each other. As the figures are now it appears that all groups differ from Mock, and that the remaining groups are not different from each other; however, the comparison between the other groups has not been performed with an appropriate test. A Dunn's or similar post-hoc test would allow for this comparison.

3) There are typo's in lines 89 and 327 where spaces are missing between words.

4) Line 262, please provide references about the only licensed H5 vaccine approved for humans.

Referee #3

(Remarks to the Author)

The authors addressed my concerns.

Response to Editor and Referee Comments

We thank the editor and Referees for their constructive feedback and comments.

While implementing the changes suggested by the Referees, we came across a small error in one of the data tables used for this manuscript. Specifically, two measured titers in an hemagglutination inhibition (HI) assay data table were not correctly converted from \log_2 values to HI titers. One data point was in the HI data underlying the antigenic map (Supplementary Table S4) and the other one underlying the antibody profiles (Supplementary Table S8). We have corrected these errors in the dataset and subsequently repeated all analysis for the resubmitted version. As expected, only small differences in the fine details of the analysis were observed. It caused no major changes in the main display items nor in any of the conclusions drawn based on the data. We have implemented the resulting changes in the manuscript accordingly. We apologize for any inconvenience that this may cause.

Please find below the Referee comments reproduced in black font, and our answer to each of them in blue font.

Referee #1 (Remarks to the Author):

In their manuscript Kok and colleagues describe a novel vaccine approach for HPAI H5N1 that is based on an antigenically centralized HA. The approach is interesting and the work is well done. However, there are several points that need the authors' attention.

Major points

1) The approach is different than previous centralized approaches which were often sequence-based and not necessarily based on antigenic data. However, the authors are pretty dismissive of these previous approaches which have actually worked well too. We thank the referee for this comment and apologize for coming across as dismissive with regards to other centralized approaches. Our intention was solely to contrast our approach with those previously taken and to highlight its novelty. As the reviewer pointed out, our approach is based on phenotypic information, which is, in our opinion, a fundamental difference with previously taken approaches. Major antigenic changes in influenza viruses are often governed by only a handful of amino acid differences in the head of the hemagglutinin (references 30-33, 35, 38-40). Centralizing diversity on genetic level will not allow these crucial differences to be considered. This reasoning served as the rational for designing an antigenically central antigen.

We agree with the referee that other approaches have shown successes. However, there are, to the best of our knowledge, no studies to date in which HI antibody breadth has

been investigated in this manner - assessing serum reactivity against over a hundred H5 viruses. Therefore, previous approaches, often using small datasets, cannot definitively demonstrate vaccine cross-reactivity.

The manuscript text has been adapted to better reflect our rationale and outline the differences between previously taken genetically centralized approaches and our antigenically centralized one (line 421-425, 435-436).

2) It would have been good to use another unrelated inactivated influenza virus as negative control, e.g. and influenza B virus preparation with adjuvant.

An unrelated influenza virus vaccine control would be relevant to address potential effect of heterosubtypic immunity, which has been done extensively by others (e.g., Nuñez et al., PMID: 38170075, Bodewes et al., PMID: 21228239, Cheng et al., PMID: 19209231). Our goal was rather to compare an improved antigenically central vaccine antigen to the current standard-of-care, i.e., a homologous vaccine to the challenge virus, and to its parent non-central wild-type antigen (Anhui_{VACC}). Additionally, the Anhui_{VACC} group serves as a negative control for protection in the absence of detectable HI antibody titers against the challenge viruses (see answer to the first comment of Referee 3). The PBS-adjuvant group (Mock_{VACC}) serves as another negative control to fully appreciate disease severity upon challenge. The text was adapted to clarify the use of the different control groups (line 259-263) and the absence of HI detectable titers in ferrets from the Anhui_{VACC} group (line 269-271).

Minor points

We thank the referee for their sharp minor points and have adapted the manuscript according to the suggestions.

1) Line 39: HA1 also contains part of the stalk.

We removed (HA1) from the sentence, line 39.

2) Line 45: In Australia too?

While no poultry outbreaks have been detected in Oceania, we refer here to the recent detection of the first human case in Australia (reference 14). While this case was imported from India, we think that it is important to highlight it to showcase the global presence of H5 GsGd viruses and their potential to spread intercontinentally, through animal and human movements.

3) Figure 1B: It would be better to label the clades within the figure, not on the side. Since not all antigens and sera of one genetic clade group together antigenically, we have decided to label the clades on the side. We think that the suggested change will decrease

the readability of Figure 1B and therefore have decided not to implement this change. In Supplementary Data S2 (https://epiv-lab.github.io/H5-antigenically-central-vaccine/Supplementary_Data_2.html), hovering over individual antigens with the cursor displays the antigen name followed by an underscore and the respective genetic clade.

4) Line 112: Are really the non-gooseGd lineages meant?

Yes, antigens (and sera) depicted in grey in Figure 1 belong to lineages other than the GsGd lineage.

5) Line 115: 'A zoomable pdf file...'

The spelling mistake was corrected.

6) Line 161 and 162: Remove 'it' twice.

The second "it" was deleted.

7) Line 181 and throughout the manuscript: It is 'alpha2.6-linked sialic acid' not 'alpha 2.6 sialic acid'. Same for 2.3 of course.

This change was adapted throughout the manuscript.

8) Line 206: Define 'GMT'.

GMT was defined previously (line 177).

9) Line 216: Please correct the strain name.

The spelling mistake was corrected.

10) Line 283: Please indicate the challenge dose for the respective viruses.

The challenge doses were indicated in the methods. For clarity, we added this information in brackets at the end of the sentence (line 304).

11) Line 283 and throughout the manuscript: Please use proper strain names or abbreviations but not 'Giza' and 'Sichuan'.

We thank the referee for this comment. We agree these abbreviations were not the best choice and have adapted these to H5N1_{Giza} and H5N6_{Sichuan}.

12) Line 300: 'trachea', not 'tracheas'

The spelling mistake was corrected.

13) Line 422, line 631: 'et al.' should be in italics.

This was corrected.

14) Line 440: The only published human studies with mRNA vaccines for avian influenza viruses (H7 and H10) do actually indicate worse immunogenicity of this platform compared to inactivated adjuvanted vaccines.

mRNA vaccines are cited in the discussion only as potential alternative platforms to more conventional approaches, along with other approaches such as vector-based ones. These platforms, enabling de novo synthesis of viral proteins, are particularly interesting for their potential to activate the cellular arm of the immune system. While the current available data about avian influenza virus mRNA vaccines may be disappointing, there is currently insufficient data to fully evaluate their potential immunogenicity in humans.

The text in the discussion has been adapted to address the referee's point and better reflect this point of view (line 461-467).

15) abbreviations in the methods section are not defined.

The missing abbreviation definitions in the methods section were added.

16) Line 510: It should be 'Dulbecco's'.

The spelling mistake was corrected.

17) Line 513 and 516: Streptomycin is typically not given in U/ml but ug/ml.

This was corrected.

18) Line 513: 'geneticin', not 'Geneticin'

This was corrected.

19) Line 617: Please describe for the reader what 'Mega10' is.

N-decanoyl-N-methylglucamine (Mega10) is used to solubilize the viral membrane proteins to produce split inactivated vaccines. This information was added, line 649-650.

20) Line 733: Please use non-capitalized letters to start words mid-sentence.

This was corrected.

21) Line 769: Where is the data from the VN assays used?

The VN assay data are depicted in Extended Data Figure 3, panel D, which is referred to in the main text line 110-112.

Referee #2 (Remarks to the Author):

Kok et al. selected 33 viruses that represented the genetic diversity of H5 influenza viruses. Using the gold-standard ferret model of influenza pathogenesis and vaccination, anti-sera was raised against these viruses and tested for cross-reactivity via hemagglutination inhibition (HI) assay to 127 different H5 antigens. After excluding 4 anti-sera and 10 antigens due to poor reactivity, the results of the cross-HI assays were used to perform antigenic cartography. This analysis revealed discordance between genetic and antigenic evolution. The antigenic map further showed that antigenic evolution did follow a specific pattern or was non-directional. From the antigenic map, several central H5 antigens were chosen for further evaluation as candidate vaccines. Mutations were introduced into the HA to alter receptor-binding preference and remove a putative glycosylation site. Ferret anti-sera was then raised against candidate vaccine antigens, and after performing cross-HI assays, the A/Anhui/1/2005 (H5N1) HA was chosen as the most suitable antigen. The stability of the A/Anhui/1/2005 HA was then evaluated during repeated passaging in eggs and sequencing of the passaged viruses revealed that the 134TA mutation was selected. This mutation was then incorporated into the final vaccine antigen. Last, to evaluate vaccine efficacy, vaccination and challenge studies were performed using the candidate vaccine and two heterologous challenge strains. Vaccines matching the challenge strains were also incorporated as comparators. After vaccination and challenge, ferrets vaccinated with the A/Anhui candidate vaccine were protected from disease and the level of protection was comparable to control vaccines matching the challenge strain. Collectively, these studies demonstrate that antigenic cartography can be used to select vaccine antigens with enhanced breadth of protection.

Comments:

The use of antigenic cartography to select a central H5 antigen represents an advancement in the design of H5 vaccines. Moreover, the experimental studies represent a large body of research, and the findings demonstrate the central H5 antigens with additional mutations induce a cross-reactive HI response. The findings are presented in detail with multiple supplemental tables and figures. The statistical analyses are appropriate and the conclusions and stated limitations are accurate.

There are sections of the manuscript that would benefit from revisions for clarity. Specifically, lines 77-93, indicate that 127 antigens were selected, and 33 anti-sera were generated based on preliminary studies. However, it would be much easier to follow if the process was instead explained as 33 anti-sera were generated and tested for cross-reactive to 127 HAs.

We thank the referee for their suggestion to clarify this part of the manuscript. However, rephrasing as suggested would not accurately recapitulate how the study was conducted. The project started with antigen selection based on diversity at the genetic level. Preliminary hemagglutination inhibition (HI) assays were performed using sera from our historical collection. Additional antigens for sera production were subsequently selected based on low cross-reactivity in HI assays. However, to accommodate the referee's comment we have adapted the text lines 78-79 and 85-86 to simplify and clarify the explanation of this process.

Line 97 indicates “four detectable titers”, is this meant to convey 4 HI units. Please clarify. We thank the referee for this comment. Detectable titers refer to HI titers which fall within the assays' detection limit and therefore have a numerical value in our merged dataset. This is in contrast to undetectable titers, which are either titers below the detection limit of the assay (for example <10) or titers set to “NA”. The latter case results from merged datapoints when the standard deviation of \log_2 transformed individual titers measured in different assays was above 1.5, suggesting the merged numerical datapoint is unreliable. We adapted the text (lines 99-104) to increase clarity. In our opinion, adding the details on the merging of the data and titer conversion in the manuscript main text would be distracting and only of interest to a few specialists, hence we provide a detailed explanation in supplementary notes 1 through 3.

Parts of the manuscript indicate the H5 antigens were engineered or rationally designed (see lines 28, 410, etc). This is slightly misleading as central antigens were chosen and mutations were introduced. The rationale for introducing the mutations is justified; however, the terms “rationally designed or engineered antigens”, at least to this reviewer, imply a completely synthetic approach rather than selecting from existing viruses.

We thank the referee and agree that using the word engineered may be misleading. However, given that our study is based on a rational approach presented and justified in the manuscript, we think that the term “rationally designed” is appropriate. We replaced the verb and adjectives “engineer” or “engineered” with “design”, “designed”, “introduced” or “mutated” depending on the context.

Extended Data Figure 1A, please clarify what is meant by “number of titers” on the y-axis.

We thank the referee for this comment and agree that this required some clarification. Hence, we have changed the y-axis label of Extended Data Figure 1A to “number of data points”. A data point refers to each unique antigen-serum combination in our dataset. This information has been included in the figure legend, and the text referring to Extended Data Figure 1A has been updated as well to accommodate this comment.

In the virus challenge studies, the authors indicate the dose used was determined in preliminary studies. Please add details to clarify if the challenge dose was lethal.

We thank the referee for this comment. As indicated (line 301-304) the doses were not chosen to be lethal, but to “induce a reproducible and consistent upper and lower respiratory tract infection”. In preliminary studies, two doses per challenge virus were tested using three ferrets per dose. The rationale was to select the lowest dose inducing a reproducible, consistent, but non-lethal, infection of both the upper and lower respiratory tract. For the H5N1_{Giza} virus challenge, doses of $10^{4.3}$ and $10^{5.0}$ TCID₅₀ per animal were tested. Both doses resulted in non-lethal infection in all ferrets at four days post-inoculation. The higher dose resulted in a consistent upper and lower respiratory infection in all ferrets, whereas the lower dose did not. Thus, the higher dose was selected. In the vaccination-challenge experiment, the back titration of the inoculum revealed a final dose of $10^{5.5}$ TCID₅₀ per animal.

For the H5N6_{Sichuan} virus challenge, doses of $10^{2.3}$ and $10^{3.7}$ TCID₅₀ per animal were tested. Both doses resulted in non-lethal infection in all ferrets at four days post-inoculation. The pneumonia observed in ferrets from the lower dose group was classified as ‘mild’, and for the higher dose group as ‘moderate’. In the higher dose group, the viral titers in the respiratory swabs and the tissue samples were more consistent between individual animals as compared to the lower dose group. For these reasons, the higher dose was selected. In the vaccination-challenge experiment, the back titration of the inoculum revealed a final dose of $10^{3.4}$ TCID₅₀ per animal. Unexpectedly, this dose was lethal to two animals in the negative control group.

Also in the methods section, please indicate the method of euthanasia.

Ferrets were euthanized through cardiac puncture, which was indicated lines 729 and 744 (methods) for the vaccination experiments but indeed omitted for the serum production. The latter was added line 704.

While the manuscript uses antigen cartography which is an advanced approach, studies such as those by Boonnak et al., (J Virology 2017) in which anti-serum to several H5 viruses was tested for cross-reactivity to select a broadly cross-reactive H5 vaccine strain should be referenced and discussed. It would be particularly useful to add details of how antigen cartography is an improvement over cross-HI alone.

We thank the referee for this comment and bringing this interesting study to our attention. We apologize for not having referenced it earlier. The selection of vaccine antigens in the study by Boonnak et al. was based on limited cross-HI data (9 antigens against 9 sera). Vaccine sera cross-reactivity was assessed only against seven heterologous antigens. Even with such a small panel, the wild-type vaccine antigens that were selected still led to some clade-specific immunity in ferrets.

As the number of data points in the cross-HI table in Boonnak et al. is significantly smaller compared to ours (81 versus 4191), this significantly smaller cross-HI dataset can be more easily interpreted without the help of antigenic cartography. However, large HI datasets, such as the one presented in our study, are essential to fully recapitulate the diversity of the antigenic properties of influenza viruses and fully appreciate cross-reactivity. As HI datasets get larger, antigenic cartography becomes critical for quantitative interpretation and ease of visualization. Additionally, antigenic cartography allows to understand antigenic relatedness between antigens based on their reactivity patterns to antisera, a parameter which is not directly measured in the HI assay. This type of clustering analyses would be extremely difficult to perform only by visual inspection of an HI table.

On the broader context, antigenic cartography has revolutionized the field of antigenic characterization of influenza viruses. Since its development 20 years ago, it has been adopted routinely to assist human influenza virus surveillance and vaccine strain selection process by the WHO (PMID: 20521635). Antigenic cartography also permits comprehensive integration of data from multiple laboratories, which would be otherwise uninterpretable. It is our expectation that the antigenic map presented in this study will be an invaluable tool to further monitor antigenic diversification of A(H5) GsGd viruses and will enhance surveillance and antigenic characterization of emerging viruses.

Finally, antigenic cartography allows not only the visualization of serum cross-reactivity in an unprecedented way through the generation of antibody landscapes (PMID: 25414313) or profiles (this study), but also a better understanding of cross-reactivity patterns. The reactivity of vaccines against newly emerging A(H5) viruses can be predicted by extrapolating their position in the antigenic map in relation to other antigens for which vaccine sera reactivity is known.

The text was refined to accommodate the referee's comment and better justify the need for antigenic cartography to interpret large cross-HI datasets (lines 89-92). Citation of the work by Boonnak et al. was added (line 435-436; reference 55).

The manuscript refers extensively to supplemental tables and figures. This is due to the complex nature of the dataset. Within the supplemental tables, it would be very helpful if vaccine and challenge strains were highlighted in bold.

We thank the referee for this comment. We have adapted the tables accordingly, and moreover, color-coded the supplementary tables to increase readability.

Within the antigenic maps, it would similarly be helpful, if the vaccine and challenge strains were denoted with symbols that made them readily identifiable from all other viruses. This would allow the reader to quickly identify these strains and assess their relationship to other viruses.

We thank the referee for this comment and pointing out the importance of allowing the reader to readily identifying the vaccine and challenge strains in the antigenic maps. To accommodate this comment, we have increased the size of the vaccine and challenge strains in Figure 3 and Supplementary Data S6-10 (see <https://epiv-lab.github.io/H5-antigenically-central-vaccine/>). Moreover, in Figure 3 we have included a display of the antigenic map similar to that from Figure 1B, color-coded by genetic clade and with the vaccine and challenge strains highlighted in bigger size. In the interactive displays of Supplementary Data 6-10, one can hover over the antigens to display their names.

Referee #3 (Remarks to the Author):

Kok and colleagues present a very nice series of studies designing and testing new H5 vaccine antigens. They used ferret antisera to generate an H5 antigenic map and then identified viral isolates that were centrally located in the antigenic map. They then modified centrally located H5 antigens by removing a glycosylation site and adding mutations that facilitate α 2-6 sialic acid binding. A split vaccine was created using one of these modified H5s—this vaccine (along with the unmodified central antigen) protected ferrets against challenge with 2 distinct H5 strains—and protection was similar to that afforded by homologous vaccines. The paper is important and presents an interesting concept, but it is not clear if the modified vaccine provides better protection compared to other mismatched antigens. Regardless, I think that this is a tremendously valuable set of experiments and it will move the field forward.

1. This is an interesting idea and the manuscript is filled with interesting and useful data. But it is not clear that the modified vaccine (AC-Anhui-VACC) is much better than the unmodified centrally located antigen (Anhui-VACC). It is not clear that this vaccine is much better than a different vaccine antigen that is not centrally located on the antigenic maps, because this type of non-centrally located antigen was not included in the vaccine challenge studies. It is possible that most of the protection in the challenge studies is due to non-neutralizing antibodies and that most H5 antigens would afford some level of protection in these experiments. All of the vaccines tested protected against Giza and there were only slightly small differences between AC-Anhui and Anhui with protection against Sichuan.

We thank the referee for this comment, giving us the opportunity to clarify the design of our study. Anhui_{VACC}, i.e. the wild-type A/Anhui/1/2005 antigen, is not centrally positioned in the antigenic map (antigen and vaccination sera (average of 12 sera) are positioned 5.05 and 3.55 AU from the center, respectively). Furthermore, the challenge viruses were chosen because their respective HAs were genetically and antigenically distinct and located at a great distance to that of Anhui_{VACC} (7.40 for H5N1_{Giza} and 10.61 AU for H5N6_{Sichuan}, respectively, outlined in line 251-254 in the manuscript). Therefore, a 'non-antigenically central antigen' suggested by the referee was actually included in the study by including Anhui_{VACC}. The text in the manuscript was modified to better highlight this aspect (lines 191, 254, 276, 430) and be more explicit about the use of the different control groups (line 259-263).

We agree with the referee that some level of protection is expected to be provided by any A(H5) immunity history, even when cross-reactivity in HI is low or inexistent. There is already a breadth of data showcasing heterosubtypic cross-immunity from past exposures with seasonal influenza A viruses (see response to Referee 1), which HAs (H1 and H3) are fully mismatched with A(H5). Such heterosubtypic immunity or cross-

immunity in the absence of HI antibodies is mediated via HA non-neutralizing, antibodies to other viral proteins like NA, NP or M2, or T-cell responses. However, the presence of cross-neutralizing antibodies is a very important and recognized correlate of protection and will improve vaccine efficacy and protection.

The goal of the two vaccination-challenge experiments was to provide a proof-of-principle of the non-inferiority of AC-Anhui_{VACC} in protection against two antigenically distinct A(H5) viruses compared to homologous vaccines. While Anhui_{VACC} provided a certain level of protection against H5N6_{Sichuan}, this vaccine remained inferior to the homologous vaccine when considering body weight loss, body temperature change and extra-respiratory viral spread, while AC-Anhui_{VACC} was non-inferior. Differences between AC-Anhui_{VACC} and Anhui_{VACC} observed upon challenge with H5N1_{Giza} were indeed less stark, yet infectious virus was detected in the cerebrum of 4/6 Anhui_{VACC} animals and in 0/6 AC-Anhui_{VACC} animals. Protecting against potential extra-respiratory spread to the brain is of the utmost importance. Taken together, we demonstrated non-inferiority of AC-Anhui_{VACC}, but not of Anhui_{VACC}, as compared to the standard of care (homologous antigen to challenge virus) against viruses located >9 AU apart from one another. This aspect was clarified lines 440-443.

Finally, the most important differences between Anhui_{VACC} and AC-Anhui_{VACC} are the contrasting breadth of the HI antibody response as observed in the antibody profiles. Given the cross-reactivity patterns of AC-Anhui_{VACC} and the results from the vaccination challenge experiments, it is expected that AC-Anhui_{VACC} will at least lead to a similar level of protection against the many viruses against which a similar or higher HI titer was measured as compared to those against the challenge viruses.

2. The authors modified the centrally located HA antigens by introducing mutations that directly altered receptor binding specificity. The authors completed experiments to document how these mutations affected specificity. It is possible that these mutations are also changing the overall avidity and HA binding strength to the turkey red blood cells used in HI assays. It is possible that these mutations decrease overall HA binding to turkey red blood cells. Viruses that bind poorer to red blood cells can have inflated HI titers, simply due to differences in binding avidity rather than differences in antigenicity. The authors should consider measuring overall avidity to determine if the higher HI values with the modified HAs is due to weaker red blood cell binding.

We agree with the referee that the cross-reactive patterns in HI of viruses with modified binding properties may be affected by avidity to red blood cells. However, our objective was not to design an antigen with high HI reactivity, but with high immunogenicity. Therefore, our analyses are not focused on the HI reactivity of the vaccine antigens, which merely serve as a screening for interesting candidates. Our conclusions that AC-Anhui_{VACC} show increased immunogenicity are based on HI assays performed with the AC-Anhui_{VACC} post-vaccination sera in combination with wild-type antigens, the vast

majority of which will present typical avian-type receptor binding properties, without any risk of inflated HI titers. The manuscript text was amended to better reflect this aspect (line 202-204).

3. Ferrets tend to produce antibodies mainly against the top of HA that target residues near the RBS. Humans typically produce antibodies that target multiple different sites, and it is possible that antigenic maps based on ferret serum are underestimating the contribution of antigen sites further from the RBS. Presumably, the loss of a glycan at HA residue 154 increases the accessibility of the top of HA, which is the immunodominant site in ferrets.

While there is to date no such information on A(H5) hemagglutinins, there are several studies showing that the reactivity of ferret sera to influenza viruses from other subtypes (A(H1), A(H2) and A(H3)) is similar to that of first-exposure human sera (Matzuzawa et al., PMID: 31652870; Koel et al., PMID: 25609810; Fonville et al., PMID: 26142433). Matsuzawa used infant sera to study the antigenic evolution of A(H2N2) viruses and showed that antigenic properties A(H2N2) viruses revealed by human sera were similar to that revealed by ferret antisera. Koel et al. showed that escape from recognition by A(H1) viruses presenting substitutions in the HA head was similar between ferret and infant sera. Fonville et al. constructed an A(H3N2) antigenic map using primary human sera and compared it to an antigenic map built from assays performed with ferret sera (Smith et al., PMID: 15218094). They showed that maps constructed with ferret and human sera were globally similar, indicating that ferret sera can be used to generate useful information on overall antigenic differences among viruses.

Data generated with primary ferret and human sera agree, suggesting that there are no major differences in terms of epitope immunodominance between ferrets and humans. However, antibody responses in humans may broaden later in time upon multiple exposure, as expected. We agree with the referee that, as concluded in the last sentence of the manuscript, the data generated in this study warrants evaluation of immunogenicity and breadth in humans.

Are the RBD and glycosylation patterns of Giza and Sichuan different, and do these viruses share homology with the modified vaccine antigen in the RBD?

The exact same N-glycosylation sites are predicted for H5N1^{Giza} and H5N6^{Sichuan}. Both H5N1^{Giza} and H5N6^{Sichuan} lack the predicted N-glycosylation site at amino acid position 154, which is also absent in AC-Anhui because of the introduced 156TA substitution. H5N1^{Giza} and H5N6^{Sichuan} differ with 38 amino acid positions in HA1, and both differ with 27 amino acid positions from AC-Anhui in HA1.

For illustration, we have included in the rebuttal a display of the A/Anhui/1/2005 HA structure highlighting the amino acid differences in HA1 between A/Anhui/1/2005 the two challenge viruses, as well as AC-Anhui, color coded as follows: amino acid positions

which only differ between A/Anhui/1/2005 and H5N1_{Giza} in red, amino acid positions which only differ between A/Anhui/1/2005 and H5N6_{Sichuan} in blue, amino acid positions which are the same in H5N1_{Giza} and H5N6_{Sichuan} but different from A/Anhui/1/2005 in purple, and positions which differ between all three HAs in green. The positions which are mutated from A/Anhui/1/2005 to AC-Anhui are highlighted in yellow, on top, position 156 (156A in AC-Anhui, H5N1_{Giza} and H5N6_{Sichuan}), on the side, position 134 (134A in AC-Anhui, H5N1_{Giza} and H5N6_{Sichuan}), 222 (unique in AC-Anhui) and 224 (unique in AC-Anhui). Given that we think that is more important to highlight in the manuscript the antigenic differences between the two challenge viruses and the vaccine antigens (distances in the antigenic map are indicated) rather than the genetic differences, we have decided to not add the display below in the manuscript and to only use it to adequately answer the question of the referee.

Does the modified vaccine elicit antibodies against the immunodominant top of HA that is similar in Giza and Sichuan?

We thank the referee for this interesting question. As put forward in the discussion, we have focused our analyses on polyclonal antibody responses. It would be very interesting to study the immune response elicited by our modified vaccine at the monoclonal level, to further understand the molecular basis for increased immunogenicity. However, the lack of reagents to perform B-cell analyses in ferrets, among other technical limitations, prevented such analyses. However, given the differences between the top of the HA of H5N1_{Giza} and H5N6_{Sichuan} and AC-Anhui, which are outlined above, we think that it is unlikely that the cross-reactivity provided by AC-Anhui against H5N1_{Giza} and H5N6_{Sichuan} only relies on similarity at the top of the HA head, e.g. 156A. Right in proximity to position 156 are located other variable positions between the three viruses, highlighted in purple, red and blue.

4. The HI tables in Supplemental tables 8 and 9 are important and should be incorporated in some manner into the main figures. I understand these data are the basis of the antigenic maps, but is there an abbreviated table or figures that can make it easy for the reader to directly compare titers (for example, direct comparison of antibodies elicited by Anhui-VACC versus AC-Anhui-VACC against individual viral strains to evaluate differences)?

We thank the referee for this comment and agree the HI data in supplemental tables 8 and 9 are important. However, given the size of these tables, we think that it is not possible to integrate them as main displays. Given that the strength of our dataset is its comprehensiveness, we think that highlighting in the main HI titers against a handful of individual viruses will be counterproductive. Instead, to improve the readability of supplemental tables 8 and 9, we have color-coded the antigen and sera names based on genetic clade and experimental group, respectively, and color-coded the cells based on HI titer and columns with the geometric mean titers per experimental group were added. This allows for a direct comparison between titers against individual strains per experimental group. Finally, one can use the interactive html displays (supplementary data S6-10), which show the antibody profiles for direct comparison of titers against individual viral antigens. The antigen names are displayed when hovering over.

5. Does 134TA lead to a glycan loss?

The 134TA substitution is not predicted to impact N-linked nor O-linked glycosylation site patterns (NetOGlyc 4.0, DTU).

We thank the editor and Referees for their constructive feedback and comments.

Please find below Referee 2's comments reproduced in black font, and our answer to each of them in blue font.

Referee #2 (Remarks to th

Kok et al. have submitted a revised version of their manuscript which seeks to use antigenic cartography to identify an antigenically central H5 HA antigen for use as a broadly H5 virus protective vaccine. The revised manuscript has addressed several previous comments; however, there are still some remaining comments to consider/address.

Comments:

1) The vaccines tested in the manuscript are adjuvanted split-virion vaccines, and the adjuvant combined with modifications to the HA are likely contributing to the breadth of the antibody response. It is unclear if the vaccines were not adjuvanted if a central H5 HA antigen would have similar protection as a homologous vaccine. In the event of a pandemic, while adjuvants would certainly be beneficial, it is unknown if they would be approved for use in humans. For the 2009 H1N1 pandemic, adjuvanted vaccines were approved in Europe but not the United States. This should be addressed either by adding rationale in the introduction or results for including an adjuvant (i.e. indicating that H5 vaccines without an adjuvant are unlikely to be immunogenic), or by listing the use of an adjuvant as a limitation in the discussion and that additional studies without an adjuvant are warranted.

There are several adjuvants that have been approved for human use by several agencies (e.g. FDA or the EMA) in combination with zoonotic influenza virus vaccines, such as AS03 or MF-59. The adjuvant used in this study is Addavax, an MF-59 like adjuvant. As the reviewer mentioned, the use of an adjuvant is inevitable when utilizing inactivated vaccines against an influenza virus the population is mostly naïve to. To clarify this point, we added this rationale in the result section, line 222-223.

2) The description of the statistical approach in the methods and figures is inconsistent, and the statistical approach requires revisions. In the methods, the authors indicate a Kruskal-Wallis test was first performed to determine statistical significance, and this was followed by pairwise Mann-Whitney U tests. There are other post-hoc tests such as a Dunn's test that are more appropriate than pairwise Mann-Whitney U tests. Mann-Whitney U tests are meant for pair-wise comparisons and do not account for multiple comparisons. If Mann-Whitney U tests are used, then the full statistical approach needs to be clarified in the figure legends. It is also not accurate to denote with a single horizontal bar that 3 or 4 groups differ from Mock. Instead, each individual comparison should be depicted, or it should be clarified in the figure legend that only the comparison between the mock and each individual group were performed and the line does not imply

the other groups are not different from each other. As the figures are now it appears that all groups differ from Mock, and that the remaining groups are not different from each other; however, the comparison between the other groups has not been performed with an appropriate test. A Dunn's or similar post-hoc test would allow for this comparison.

We thank the reviewer for this comment. We did perform the Benjamin-Hochberg multiple comparison correction following the paired two-tailed Mann Whitney U tests but unfortunately omitted to mention it in the Methods. We apologize for the inconvenience.

However, we agree with the reviewer that Dunn's test, upon multiple comparison correction, is a valuable alternative. It preserves the ranking of the Kruskal-Wallis test, while Mann Whitney tests are agnostic to Kruskal-Wallis results. Therefore, we have revised our statistics approach. In the revised manuscript, we assessed statistical significance using a Kruskal-Wallis test followed by pairwise two-sided Dunn's tests with Bonferroni correction for multiple comparisons. All pairwise comparisons were tested, and p values were adjusted for multiple comparisons.

As indicated in each legend, only statistically significant differences that are shown, with the corresponding exact p value, to avoid overcrowding of the figures.

No major differences were observed when using the revised method compared to the original one. In the vast majority of the cases, significance between Anhui_{VACC} and Mock_{VACC} was lost, exemplifying more clearly the superiority of AC-Anhui_{VACC} over Anhui_{VACC}. In three instances for the H5N1_{Giza} challenge - bodyweight loss, relative lung weight and temperature - significance between Giza_{VACC} or AC-Anhui_{VACC} and Mock_{VACC} was lost. Manuscript text was amended to reflect these differences lines 264, 274, 283-288, 293-296.

3) There are typo's in lines 89 and 327 where spaces are missing between words.

We thank the reviewer for noticing these errors, which were corrected in the revised manuscript.

4) Line 262, please provide references about the only licensed H5 vaccine approved for humans.

We added the following reference: [PMID: 40090339](https://pubmed.ncbi.nlm.nih.gov/40090339/).